# Revealing the tissue-level complexity of endogenous glucagon-like peptide-1 receptor expression and signaling

Julia Ast [1], Daniela Nasteska[1], Nicholas H. F. Fine [1], Daniel J. Nieves [2], Zsombor Koszegi [1], Yann Lanoiselée[1], Federica Cuozzo [1], Katrina Viloria[1], Andrea Bacon[3], Nguyet T. Luu[4,5], Philip N. Newsome[4,5], Davide Calebiro [1], Dylan M. Owen [2], Johannes Broichhagen [6] ✉ & David J. Hodson [1,7] ✉

The glucagon-like peptide-1 receptor (GLP1R) is a class B G protein-coupled receptor (GPCR) involved in glucose homeostasis and food intake. GLP1R agonists (GLP1RA) are widely used in the treatment of diabetes and obesity, yet visualizing the endogenous localization, organization and dynamics of a GPCR has so far remained out of reach. In the present study, we generate mice harboring an enzyme self-label genome-edited into the endogenous *Glp1r* locus. We also rationally design and test various fluorescent dyes, spanning cyan to far-red wavelengths, for labeling performance in tissue. By combining these technologies, we show that endogenous GLP1R can be specifically and sensitively detected in primary tissue using multiple colors. Longitudinal analysis of GLP1R dynamics reveals heterogeneous recruitment of neighboring cell subpopulations into signaling and trafficking, with differences observed between GLP1RA classes and dual agonists. At the nanoscopic level, GLP1Rs are found to possess higher organization, undergoing GLP1RA-dependent membrane diffusion. Together, these results show the utility of enzyme self-labels for visualization and interrogation of endogenous proteins, and provide insight into the biology of a class B GPCR in primary cells and tissue.

G protein-coupled receptors (GPCRs) are one of the most diverse classes of membrane proteins, with proven therapeutic potential[1,2]. Most of our understanding concerning GPCR signaling and localization is derived from experiments in heterologous cell lines, or static snapshots of immuno-stained tissue. While cell lines have informed (sub)cellular pharmacology, they do not allow endogenous GPCRs to be interrogated, nor do they replicate the tissue environment where heterogeneity and cell-cell interactions are critical for shaping signaling responses[3–6]. Similarly, immunostaining is fraught with poorly validated and non-specific reagents, and does not allow GPCRs to be dynamically followed in time and space within the same cell(s)[7]. Thus, we are still missing key information about GPCR localization and function, which is holding back mechanistic understanding. However, visualizing and interrogating

[1]Institute of Metabolism and Systems Research (IMSR), and Centre of Membrane Proteins and Receptors (COMPARE), University of Birmingham, Birmingham, UK. [2]Institute for Immunology and Immunotherapy, and Centre of Membrane Proteins and Receptors (COMPARE), University of Birmingham, Birmingham, UK. [3]Genome Editing Facility, Technology Hub, University of Birmingham, Birmingham, UK. [4]National Institute for Health Research Biomedical Research Centre at University Hospitals Birmingham NHS Foundation Trust, University of Birmingham, Birmingham, UK. [5]Centre for Liver and Gastrointestinal Research, Institute of Immunology and Immunotherapy, University of Birmingham, Birmingham, UK. [6]Leibniz-Forschungsinstitut für Molekulare Pharmakologie, Berlin, Germany. [7]Oxford Centre for Diabetes, Endocrinology and Metabolism (OCDEM), NIHR Oxford Biomedical Research Centre, Churchill Hospital, Radcliffe Department of Medicine, University of Oxford, Oxford OX3 7LE, UK. ✉e-mail: broichhagen@fmp-berlin.de; david.hodson@ocdem.ox.ac.uk

endogenous GPCRs in the native tissue context remains a high bar to achieve.

These challenges are epitomized by the glucagon-like peptide-1 receptor (GLP1R), a prototypical class B GPCR involved in the regulation of glucose homeostasis, food intake, inflammation and hypertension[7-9]. To date, pharmacological studies of GLP1R have focused on transiently- or stably-transfected immortalized cell systems, including CHO, HEK293 and INS1 832/13[10-13]. Visualization of GLP1R is further hampered by its relatively low expression levels and lack of specific antibodies[14,15], leading to mis-assignment of numerous targets, for example in the liver (reviewed in refs. [7,16]). In addition, the known specific antibodies do not work reliably in GLP1R agonist (GLP1RA) target tissues such as the adult brain using fluorescent immunohistochemistry[16], limiting mechanistic investigation in preclinical models.

To circumvent these and other issues, we recently developed LUXendins, antagonist peptidic probes that bind the GLP1R orthosteric site[17,18]. LUXendins provided the first snapshot into the tissue-level regulation of GLP1R, showing the existence of nanodomains in pancreatic beta cells and brain[17]. However, using LUXendins, GLP1R can only be studied in the non-stimulated state, precluding examination of different receptor pools or responses to clinically relevant GLP1RA. While fluorescent congeners exist for most GLP1RA, they do not allow GLP1R dynamics to be assessed before and after stimulation[19-22]. Glp1r-Cre reporter mice also exist although are unable to measure endogenous GLP1R distribution in space and time[23,24].

To allow GLP1R to be precisely visualized and interrogated, we set out to CRISPR engineer a mouse model in which a SNAP-tag enzyme self-label is knocked into the *N*-terminus, between the signal peptide and ectodomain of the GLP1R. Alongside the mouse, a range of small molecule fluorescent dyes were rationally designed and systematically tested for their ability to effectively label SNAP-tagged GLP1R in complex tissue. By combining these technologies, we show that endogenous GLP1R can be efficiently and specifically detected using multiple colors, without influencing orthosteric binding and activity. Further, we provide insight into multicellular GLP1R signaling, including agonist-dependent heterogeneous recruitment of cells into trafficking.

## Results

### Generation and phenotyping of GLP1R$^{SNAP/SNAP}$ mice

SNAP-tags are well-suited to GPCR biology, since they are small (<20 kDa), minimally interfere with GPCR signaling and trafficking, and allow different receptor pools to be labeled with distinct fluorophores[25,26] (Fig. 1a). Notably, the SNAP-GLP1R construct is widely used for cell transfection experiments and is well characterized in terms of signaling/trafficking by multiple investigators. In addition, the SNAP-tag has been employed for tissue staining in *Drosophila*, mouse and pig, albeit using transgenic approaches[26-30]. While genetically-encoded fluorophores are also applicable for tagging GPCRs, they are less flexible and preclude a number of imaging modalities e.g. stimulated emission depletion nanoscopy (STED) and stochastic optical reconstruction microscopy (STORM). To this end, CRISPR/Cas9 genome editing was used to knock-in the SNAP$_f$-tag after the N-terminal signal sequence of the GLP1R. Thus, SNAP labeling is able to faithfully report endogenous GLP1R levels. The construct was initially tested in vitro and found to signal identically to human GLP1R-GFP and SNAP-GLP1R constructs widely used in cell biology applications (Fig. 1b). $EC_{50}$ values (Exendin4 induced cAMP) were as follows: SNAP$_f$-mGLP1R = 4.22 pM (CI 2.93 pM–6.07 pM), SNAP-hGLP1R = 8.02 pM (CI 6.20 pM–10.4 pM), hGLP1R-GFP = 5.00 pM (CI 3.94–6.40 pM) (Fig. 1b). We note that mouse and human GLP1R cannot be differentiated by their $EC_{50}$'s[31], providing high confidence about the integrity of our construct. Demonstrating functionality of the

enzyme self-label, the SNAP$_f$-mGLP1R construct was readily labeled using BG-TMR (Fig. 1c).

As a start, single-guide RNA (sgRNA) targeting exon 1 of *Glp1r* and a repair template encoding the SNAP$_f$-tag were injected into the pronucleus of embryos from Cas9-overexpressing mice (Fig. 1d). Two offspring integrated repair template (Fig. 1e) and one founder was taken forward for breeding. Following 1-2 generations of back-crossing to C57BL6/J mice, off-target mutations could no longer be detected in predicted loci (Supplementary Fig. 1). GLP1R$^{SNAP/SNAP}$ and GLP1R$^{WT/WT}$ littermates were phenotypically indistinguishable, including for body weight from 4–8 weeks (Fig. 1f, g), as well as oral glucose tolerance, determined largely by the incretin effect (Fig. 1h, i). No differences were observed between males and females in this regard (Fig. 1f–i) and as such in vitro studies used tissue from both sexes.

GLP1R$^{WT/WT}$ and GLP1R$^{SNAP/SNAP}$ islets could be readily labeled with the fluorescent GLP1R antagonist LUXendin645, showing intact GLP1R binding (Fig. 1j). GLP1R expression (Fig. 1k) was similar in GLP1R$^{WT/WT}$ and GLP1R$^{SNAP/SNAP}$ islets, as assessed using a GLP1R monoclonal antibody (mAb), rigorously validated in GLP1R$^{-/-}$ tissue by us and others (reviewed in ref. [32]). Note that mAb signal was detected in both surface and intracellular GLP1R pools, since tissue needs to be permeabilized for successful antibody staining. As expected, the GLP1RA Exendin4(1-39) (Exendin4; Ex4) was able to stimulate similar magnitude insulin secretory (Fig. 1l) and cAMP responses in GLP1R$^{SNAP/SNAP}$ and GLP1R$^{WT/WT}$ islets (Fig. 1m–p). Together, these results confirm that the N-terminal SNAP-tag knock-in does not influence endogenous GLP1R signaling, paving the way for functional in vitro studies in primary tissue.

### Labeling SNAP-tagged GLP1R in complex tissue

We used pancreatic islets as an exemplar GLP1R-expressing tissue with known cell-type distributions[17,23,24,33]. SNAP-labeling is highly flexible and can be achieved either pre- or post-fixation, allowing imaging of live and fixed tissue. Previous studies have shown the utility of SNAP-tag for labeling tissue in vivo[28,34]. However, labeling of endogenously expressed protein has yet to be achieved, and head-to-head comparison of dyes is lacking. We thus tested several SNAP-tag dyes, spanning cyan through far-red spectra (~510–680 nm) and with different photophysical properties (Fig. 2a, b). To empirically determine the optimal dye for SNAP-tag labeling in the tissue setting, we initially tested the following $O^6$-benzylguanine (BG) linked fluorophores: Oregon Green (OG), carbopyronine (CPY), tetramethylrhodamine (TMR), Janelia-Fluor549 (JF$_{549}$), Cyanine5 (Cy5), silicon rhodamine (SiR) and JaneliaFluor646 (JF$_{646}$) that display different photophysical properties (i.e. excitation/emission wavelength, quantum yield and extinction coefficient). All of these dyes are characterized by their cell permeability as well as fluorogenicity upon SNAP-tag binding.

Dye performance in GLP1R$^{SNAP/SNAP}$ islets was assessed using mean fluorescence intensity along a line-scan passing from cytoplasm -> membrane -> cytoplasm. BG-OG labeled membranes in GLP1R$^{SNAP/SNAP}$ islets following 1 h incubation. However, cytoplasmic accumulation was also noticed in GLP1R$^{WT/WT}$ islets, demonstrating some non-specific GLP1R-independent uptake of the dye (Fig. 2b, c) (Supplementary Fig. 2a). BG-CPY was unable to specifically label GLP1R$^{SNAP/SNAP}$ islets, showing equivalent staining in GLP1R$^{WT/WT}$ islets (Fig. 2b, c) (Supplementary Fig. 2a). Both BG-TMR and BG-JF$_{549}$ displayed bright membrane-localized signal, although a stronger signal was observed for the fluorogenic JF$_{549}$ dye, as shown by the narrower fluorescence peaks around the cell membrane (Fig. 2b, c) (Supplementary Fig. 2a). A similar tendency was observed for the far-red dyes: BG-JF$_{646}$ was more specific than BG-SiR, while BG-Cy5 was unable to specifically label GLP1R$^{SNAP/SNAP}$ islets (Fig. 2b, c) (Supplementary Fig. 2a).

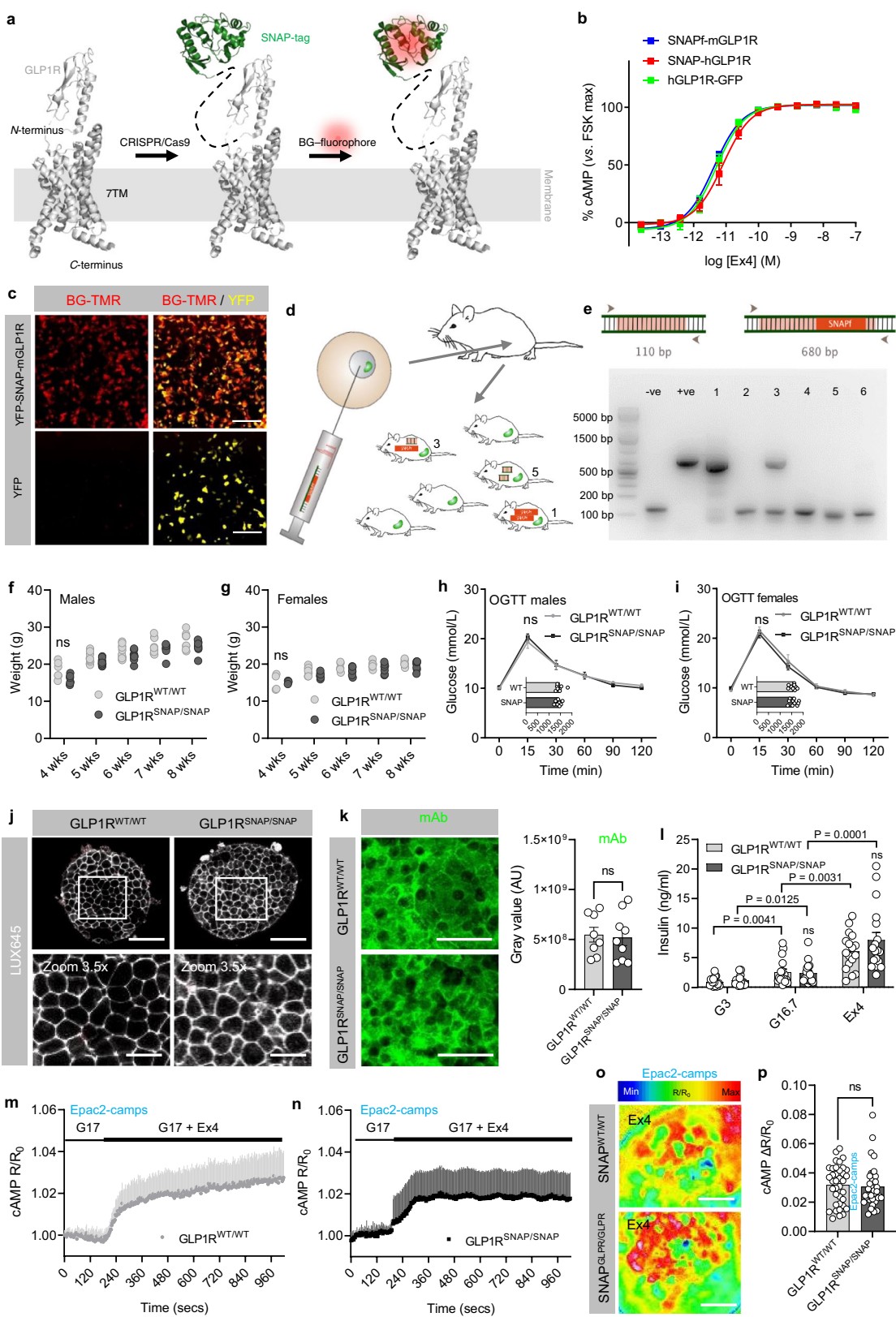

Demonstrating functionality of the endogenous SNAP-tag, BG-TMR labeling could be blocked by prior application of BG-Block, a non-fluorescent SNAP label (Fig. 2d) (IntDen: vehicle = $5.48 \times 10^6 \pm 2.29 \times 10^6$ gray value, BG-TMR = $4.86 \times 10^7 \pm 2.47 \times 10^7$ gray value, BG-TMR + BG-Block = $1.69 \times 10^7 \pm 1.22 \times 10^7$ gray value; mean ± SD, P < 0.0001, one-way ANOVA with Bonferonni's post-hoc test; F = 160.3, DF = 2) (n = 21–26

islets, 6 animals). Labeling specificity was shown in live islets by co-staining with LUXendin645 (Fig. 2e) (Supplementary Fig. 2b), as well as in fixed islets using monoclonal antibody (mAb) against GLP1R (Fig. 2f, g) (Mander's co-efficient for overlap with mAb: BG-TMR = $0.958 \pm 0.037$, BG-JF$_{549}$ = $0.970 \pm 0.064$; mean ± S.D.) (BG-TMR: n = 23 islets, 10 animals; BG-JF$_{549}$: n = 20 islets, 9 animals). Together, these data

**Fig. 1 | Generation and validation of GLP1R^SNAP/SNAP mice. a** SNAP-tags react with benzylguanine (BG)-linked substrates, allowing fluorophore labeling. **b** N-terminal SNAP_f does not influence potency of mouse (m)GLP1R cAMP generation versus SNAP-human(h)GLP1R and hGLP1R-GFP constructs ($n = 3$ replicates). **c** BG-TMR labels YFP-SNAP_mGLP1R-AD293 cells (scale bar = 205 μm) (representative images from $n = 3$ replicates). **d, e** Pronuclear injection of sgRNA and repair template into fertilized Cas9-overexpressing oocytes (**d**) leads to knock-in of the SNAP_f-tag at the *Glp1r* locus (WT: 110 bp vs. SNAP_mGLP1R: 680 bp) (**e**) (uncropped agarose gel shows repair template integration in two out of six offspring). **f, g** Body weight in 4–8 week old male (**f**) and female (**g**) GLP1R^WT/WT and GLP1R^SNAP/SNAP mice (two-way ANOVA with Bonferroni's test; male F = 0.62, DF = 4; female $F = 0.33$, DF = 4) ($n = 9$ male mice; 7 female mice). **h, i** Oral glucose tolerance in male (**h**) and female (**i**) GLP1R^WT/WT and GLP1R^SNAP/SNAP mice (area under the curve is inset) (two-way RM ANOVA with Bonferroni's test; male $F = 0.50$, DF = 5; female $F = 0.81$, DF = 5) ($n = 9$

male mice; 7 female mice). **j** LUXendin645 binds GLP1R^WT/WT and GLP1R^SNAP/SNAP islets ($n = 6$ islets, 3 animals) (scale bar = 85 μm, zoom-in = 24 μm). **k** GLP1R expression is similar in islets isolated from GLP1R^WT/WT and GLP1R^SNAP/SNAP mice ($n = 30$ islets, 6 animals) (IntDen; integrated density) (scale bar = 42.5 μm). **l** Glucose- and Exendin4 (Ex4)-stimulated insulin secretion are similar in islets isolated from GLP1R^WT/WT and GLP1R^SNAP/SNAP mice (comparison within genotype: one-way RM ANOVA with Bonferroni's test; F (GLP1R^WT/WT) = 24.33, $F$ (GLP1R^SNAP/SNAP) = 28.46, DF = 2) (comparison between genotype: two-way ANOVA with Bonferroni's test; $F = 1.33$, DF = 2) ($n = 17$ replicates, 3 animals). **m–p** Ex4-stimulated cAMP rises are similar between GLP1R^WT/WT (**m**) and GLP1R^SNAP/SNAP (**n**) mice, shown by representative images (**o**) and peak intensity between frames 120–360 (**p**) ($n = 3$ replicates, 5 animals) (two-sided unpaired *t*-test). Scale bar = 30 μm. **P < 0.05, **P < 0.01, NS, non-significant. Bar and line graphs show individual datapoints and mean ± SEM. Source data are provided as a Source Data file.

show that labeling of endogenous GPCR can be achieved in tissue, that TMR and JF_549 dyes perform best for labeling of surface + intracellular (i.e. total) GLP1R, and that labeling is highly specific.

## Surface-labeling SNAP-tagged GLP1R in complex tissue

Sulfonation renders dyes cell-impermeable, allowing surface-labeling of GPCRs[35]. We wondered whether such dyes would be advantageous in complex tissue by restricting labeling to the cell surface, where the majority of GLP1R is present in its non-stimulated state, thus increasing signal fidelity. Two different sulfonation strategies were employed depending upon the dye modified: addition of a sulfonate to the benzylguanine (SBG-TMR/SiR)[26], or via a handle added to the JF_549/JF_646 fluorophore (Sulfo549 and Sulfo646, respectively)[35] (Fig. 3a). Notably, SBG-TMR and BG-Sulfo549 greatly improved labeling of GLP1R^SNAP/SNAP islets, with similar performance seen for both dyes, as shown by representative images and peak intensity line scans (Fig. 3b, d). Similarly, SBG-SiR and BG-Sulfo646 outperformed their non-sulfonated counterparts (Fig. 3c, d). Notably, strong GLP1R^SNAP/SNAP labeling was still present within the islet core (50 μm; ~5 cell layers), assessed using confocal z-stacks of BG-Sulfo549 signal (Supplementary Fig. 3a, b). Suggesting that the sulfonation strategy works across cell preparations, SBG-TMR labeled intact and dissociated beta cells (Supplementary Fig. 4a). Further demonstrating specificity, labeling was prevented by prior application of BG-Block to occupy the SNAP binding site (IntDen: vehicle = $2.14 \times 10^7 \pm 2.22 \times 10^7$ gray value, BG-Sulfo549 = $1.09 \times 10^8 \pm 1.02 \times 10^8$ gray value, BG-Sulfo549 + BG-Block = $4.34 \times 10^7 \pm 3.76 \times 10^7$ gray value; mean ± SD, $P < 0.0001$, one-way ANOVA with Bonferonni's post-hoc test; $F = 14.68$, DF = 2) ($n = 25$–33 islets, 5 animals) (Supplementary Fig. 4b).

Remarkably, the sulfonation strategy also worked for Cy5-based dyes (Fig. 4a). Unmodified BG-Cy5 displayed non-specific labeling in GLP1R^WT/WT and GLP1R^SNAP/SNAP islets (Fig. 4b). While SBG-Cy5 labeling was membrane-localized in GLP1R^SNAP/SNAP islets, some non-specific uptake could be detected in cells at the periphery of GLP1R^WT/WT islets (Fig. 4b). By contrast, SulfoCy5 labeling was highly specific for GLP1R^SNAP/SNAP islets, with almost no detectable signal in GLP1R^WT/WT islets, suggesting that the improvements in labeling were not only due to stronger SNAP-binding, but also a decrease in non-specific accumulation into the cell cytoplasm (Fig. 4b). By contrast, BG-Alexa647, whose molecular scaffold is highly sulfonated, was unable to label GLP1R^SNAP/SNAP islets, suggesting that side-chain sulfonation leads to too much target repulsion in tissue (Fig. 4b). Demonstrating specificity, BG-SulfoCy5 labeling was strongly co-localized with GLP1R mAb (Fig. 4c) (Mander's co-efficient = 0.932 ± 0.044; mean ± S.D.), as well as LUXendin551 staining (Fig. 4d) ($n = 13$–22 islets, 5–10 animals). Lastly, BG-SulfoCy5 labeling could be prevented by pre-incubation with BG-Block (IntDen: vehicle = $1.11 \times 10^7 \pm 1.44 \times 10^7$ gray value, BG-SulfoCy5 = $2.87 \times 10^8 \pm 2.20 \times 10^8$ gray value, BG-SulfoCy5 + BG-Block = $7.98 \times 10^6 \pm 6.15 \times 10^6$ gray value; mean ± SD, $P < 0.05$,

one-way ANOVA with Bonferonni's post-hoc test; $F = 3.88$, DF = 2) ($n = 7$–8 islets, 3 animals) (Fig. 4e, f).

Together, these data show that sulfonated SNAP-dyes label the cell surface, with particular application for the visualization of endogenous proteins in complex tissue. The data also caution against the use of Alexa647 - an exemplar direct STORM (*d*STORM) fluorophore - in live complex tissue/in vivo, and suggest that SulfoCy5 might be a better alternative.

## Multicellular regulation of GLP1R trafficking/internalization

Having established the validity of GLP1R^SNAP/SNAP mice, as well as tested a range of SNAP labels for their performance in complex tissue, we next set out to interrogate endogenous GLP1R biology. Insulin-secreting beta cells are the predominant cell type that expresses GLP1R in the pancreatic islets[17,23,33]. Of note, heterogeneous beta cell states have been characterized, spanning differences in insulin secretion, maturity, proliferation and activity[3–6,36]. Moreover, beta cells within the islet are electrically-coupled, as well as regulated by numerous paracrine signals[4,37]. We therefore reasoned that GLP1R signaling might be similarly complex.

To examine this, GLP1R^SNAP/SNAP islets were incubated with SBG-TMR before 3D cell-resolution time-lapse acquisitions of GLP1R internalization at 0, 10, 20 and 30 min post-ligand stimulation (Fig. 5a). In response to vehicle, ~30% of cells across the imaged population showed GLP1R internalization (Fig. 5a–c), quantified by the appearance of distinct fluorescent puncta. Suggesting that the observed internalization reflected GLP1R constitutive signaling, the antagonist Exendin4(9–39) (Ex9) tended to decrease cell recruitment (Fig. 5a–c) (29.6 ± 12.1% versus 16.5 ± 6.6%, vehicle versus Ex9 at 30 min; $P = 0.0802$, unpaired *t*-test) ($n = 5$–9 islets, 3–5 animals). By contrast, the potent GLP1RA Ex4 led to widespread GLP1R internalization, with 70–80% of GLP1R-expressing cells recruited into the process during the 30 min imaging period (Fig. 5a–c) (69.4 ± 13.2% versus 29.6 ± 12.1%, Ex4 versus vehicle at 30 min; $P < 0.042$, unpaired *t*-test) ($n = 8$–9 islets, 3–5 animals).

We next wondered whether other GLP1RA might exert similar effects on GLP1R trafficking/internalization. Semaglutide, a third generation GLP1RA with potent anorectic and insulinotropic properties, has been shown to possess similar signaling properties to Ex4 in stably-transfected GLP1R-expressing cell lines[38]. However, in islets, semaglutide was only able to recruit ~50% cells into GLP1R trafficking over time (Fig. 5a–c), significantly lower than Ex4 (69.4 ± 13.2% versus 44.6 ± 19.8%, Ex4 versus semaglutide at 30 min; $P = 0.0327$, unpaired *t*-test) ($n = 4$–8 islets, 3 animals). Similar results were seen for the dual GLP1R/GIPR agonist tirzepatide (Fig. 5a–c) (69.4 ± 13.2% versus 43.8 ± 16.0%, Ex4 versus tirzepatide at 30 min; $P = 0.0089$, unpaired *t*-test) ($n = 7$–8 islets, 3 animals). Further time-lapse analysis showed that all ligands increased GLP1R internalization strength - measured as the normalized intensity of GLP1R puncta in the cell - *versus*

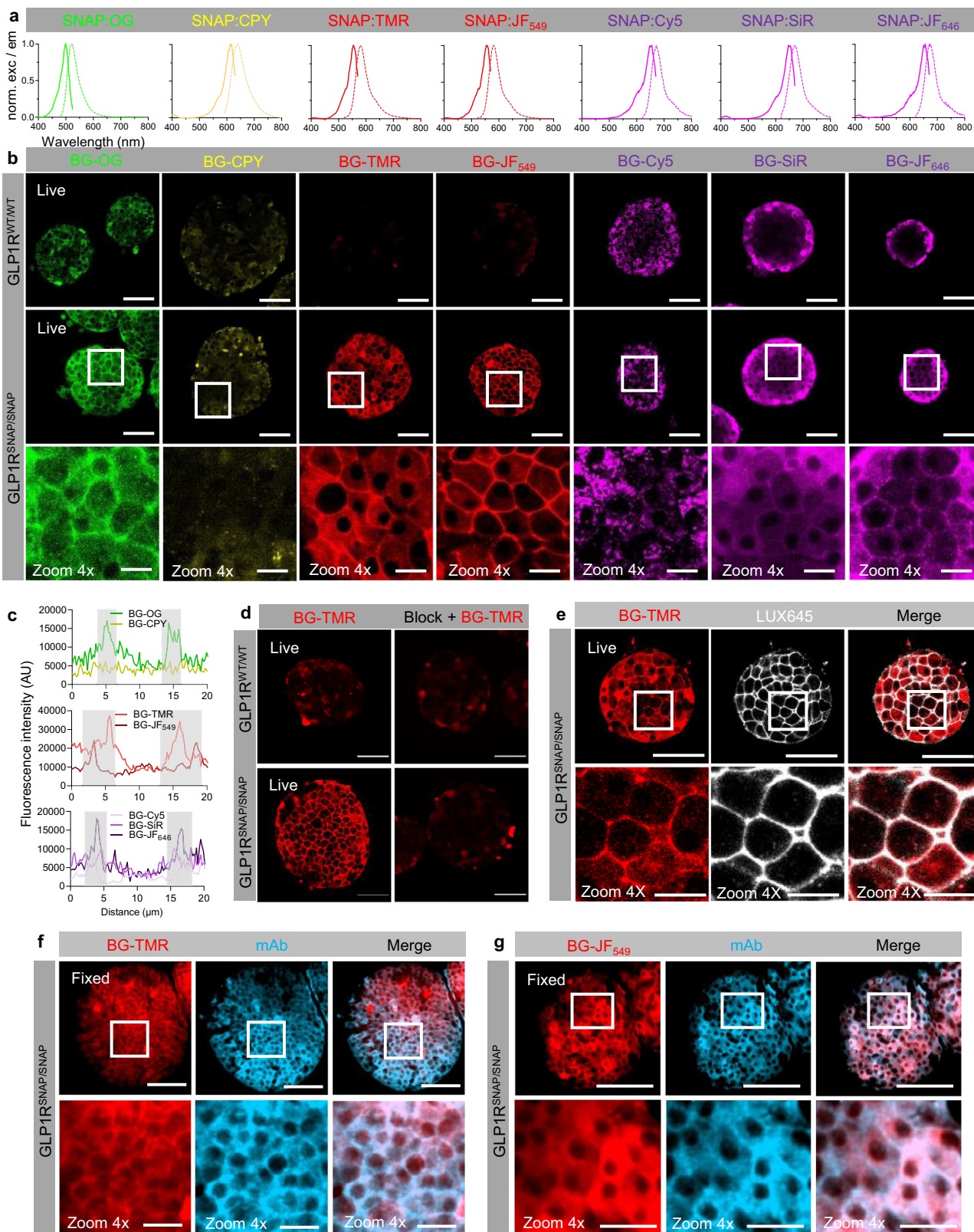

vehicle and Ex9, with Ex4 outperforming semaglutide and tirzepatide (Fig. 5d) (cells with internalization strength >1.2 at 30 min: vehicle = 20.5 ± 13.6%, Ex9 = 15.9 ± 10.8%, Ex4 = 58.8 ± 13.1%, semaglutide = 28.6 ± 13.1%, tirzepatide = 30.0 ± 11.9%; mean ± S.D, P = 0.03–0.0003, Student's $t$-test) ($n$ = 4–8 islets, 3–5 animals). Identical trends were observed for internalization rate, calculated as

the build-up of cytoplasmic fluorescence intensity over time (mean internalization rate: vehicle = 85.8 ± 76.1 min$^{-1}$, Ex4 = 213.0 ± 140.5 min$^{-1}$, Ex9 = 89.2 ± 104.1 min$^{-1}$, semaglutide = 126.6 ± 101.5 min$^{-1}$, tirzepatide = 135.4 ± 102.5 min$^{-1}$; mean ± S.D., $P < 0.0001$, one-way ANOVA, Bonferonni's post-hoc test; $F = 41.68$, DF = 4) ($n$ = 4–8 islets, 3–5 animals) (Fig. 5e).

**Fig. 2 | Rationale comparison of SNAP labels for visualization of endogenous targets in complex tissue. a** Excitation and emission spectra of cyan to far-red dyes tested for tissue labeling. **b** BG-OG labels GLP1R$^{SNAP/SNAP}$ islets, although some non-specific cytoplasmic accumulation is apparent ($n = 12$ islets, 4 animals). BG-CPY and BG-Cy5 are unable to specifically label GLP1R$^{SNAP/SNAP}$ islets ($n = 13$ islets, 4 animals for BG-CPY, $n = 15$ islets, 6 animals for BG-Cy5). BG-TMR and BG-JF$_{549}$ lead to bright and specific labeling of GLP1R$^{SNAP/SNAP}$ islets ($n = 23$ islets, 4 animals for BG-TMR, $n = 14$ islets, 3 animals for BG-JF$_{549}$). BG-SiR and BG-JF$_{646}$ leads to membrane labeling in GLP1R$^{SNAP/SNAP}$ islets, although some non-specific accumulation is seen at the tissue fringe ($n = 10$ islets, 4 animals for BG-SiR, $n = 10$ islets, 3 animals for BG-

JF$_{646}$). **c** Representative line profiles showing membrane labeling by the various SNAP labels in GLP1R$^{SNAP/SNAP}$ islets (gray shaded area shows membranous margins either side of the cell). **d** Pre-incubation of GLP1R$^{SNAP/SNAP}$ islets with SNAP-Block prevents subsequent labeling with BG-TMR ($n = 45$ islets, $n = 6$ animals). **e** SNAP labeling with BG-TMR co-localizes with orthosteric labeling using LUXendin645 (LUX645) ($n = 17$ islets, 4 animals). **f, g** BG-TMR (**f**) and BG-JF$_{549}$ (**g**) labeling co-localizes with GLP1R mAb staining ($n = 20$ islets, 9 animals). Scale bar = 85 μm for all images, except zoom in scale bar = 17 μm. Source data are provided as a Source Data file.

### GLP1R trafficking/internalization is a heterogenous cell trait

Following stimulation with Ex4, only 70% of cells were found to undergo GLP1R trafficking/internalization, suggesting that not all cells are destined toward activation at the tissue level (Fig. 5b). We therefore investigated how GLP1R trafficking/internalization was distributed across the beta cell population. In response to all GLP1RA, cells were found to activate with different delays, leading to apparent recruitment of cells into GLP1R trafficking/internalization (Fig. 6a–e). From this, different subpopulations of beta cells could be identified according to the sequence of their activation. Notably, differences could be observed between GLP1RA/dual agonist versus control (vehicle and Ex9). Ex4 activated all subpopulations equally (Fig. 6c, h, m, p), with a similar number of cells recruited into trafficking/internalization at 10, 20 and 30 min timepoints (cumulative cell number: 10 min = 10.8 ± 6.4 cells; 20 min = 3.3 ± 2.2 cells; 30 min = 4.0 ± 2.9 cells; P = 0.0006–0.0349, RM one-way ANOVA, Dunnett's post-hoc test; F = 16.60, DF = 3) ($n = 8$ islets, 3 animals). Compared to Ex4, semaglutide and tirzepatide predominantly activated an early subpopulation (cumulative cell number at 20 min: vehicle = 3.4 ± 3.1 cells, Ex9 = 5.2 ± 3.6 cells, Ex4 = 14.3 ± 5.4 cells, semaglutide = 3.3 ± 2.2 cells, tirzepatide = 9.0 ± 8.5 cells; mean ± S.D., P = 0.029–0.038, one-way ANOVA, Bonferonni's post-hoc test; F = 3.20, DF = 4) ($n = 4$–8 islets, 3–5 animals) (Fig. 6d, e, i, j, n–p). The effects of the various agonists upon subpopulation trafficking are summarized in Fig. 6p. Together, these data demonstrate a higher order of GLP1R signaling in tissue, with heterogeneous cell states emerging with shared internalization patterns, strength and rate.

### Higher organization of endogenous GLP1R

To quantify higher organization of endogenous GLP1R, we subjected intact GLP1R$^{SNAP/SNAP}$ islets to *d*STORM. Since *d*STORM requires photoblinking for signal localization, we tested BG-SulfoCy5 for labeling capability in fixed tissue. Following incubation with BG-SulfoCy5, washing three times and formalin-fixation, signal could still be detected in GLP1R$^{SNAP/SNAP}$, but not GLP1R$^{WT/WT}$ islets using confocal microscopy (IntDen: GLP1R$^{SNAP/SNAP}$ = 6.78 × 10$^6$ ± 1.81 × 10$^6$ gray value, GLP1R$^{WT/WT}$ = 2.57 × 10$^6$ ± 2.94 × 10$^6$ gray value; mean ± S.D., P < 0.0175, Mann-Whitney test) ($n = 7$ islets, 3–5 animals) (Fig. 7a). We thus progressed to *d*STORM nanoscopy to quantify endogenous GLP1R at the cell surface with ~50 nm lateral resolution. Confirming specific localization of signal from SNAP-GLP1R, no signal could be detected in BG-SulfoCy5-treated GLP1R$^{WT/WT}$ islets (Fig. 7b). In GLP1R$^{SNAP/SNAP}$ islets, *d*STORM revealed the higher organization of GLP1R at the single molecule level with unprecedented detail. Individual GLP1R and clusters thereof could be readily detected at the cell surface (Fig. 7c). To understand whether the organization of GLP1R at the membrane was random or ordered, we performed density-based spatial clustering appearance with noise (DBSCAN) cluster analysis (Fig. 7d). Since this algorithm separates signals that are in close proximity to each other with respect to lone outliers, we confirmed nanodomain arrangement of SNAP-GLP1R in mouse beta cells (GLP1R$^{SNAP/SNAP}$ = 1.41 ± 0.94 clusters/μm$^2$, GLP1R$^{WT/WT}$ = 0.12 ± 0.23 clusters/μm$^2$; mean ± S.D.; P = 0.006, Mann-Whitney test) ($n = 4$–8 islets, 3 animals).

To quantify the activation dynamics of GLP1R at the cell surface, dissociated cells were titrated with SNAP-Surface549 to resolve individual receptors for single-molecule localization microscopy (SMLM) (Fig. 7e). SNAP-Surface549 was used here to allow cross-comparison and benchmarking with previously published SMLM data[39–41]. In their resting, non-stimulated state, a bimodal distribution of GLP1R trajectories was observed encompassing freely-diffusing and trapped populations, as shown by the presence of two peaks in the diffusion-frequency plot (Fig. 7f, g). Following ligand-stimulation with Ex4, a single population of GLP1R emerged, which was poorly diffusive and trapped at the membrane, presumably due to G protein engagement and downstream signaling (Fig. 7f, g). Thus, ligand-dependent membrane diffusion arrest is a key component of endogenous GLP1R signaling in primary cells.

## Discussion

Tissue-level detection of GPCRs, including the class B GLP1R, has so far relied on antibodies or fluorescent ligands, precluding their dynamic and longitudinal assessment in non-stimulated and stimulated states. We thus generated mice in which a SNAP-tag was knocked-in between the signal peptide and the ectodomain of the GLP1R, affording enzyme self-labeling of an endogenous GPCR whilst leaving the orthosteric site crucial for signaling untouched. We also tested a color-palette of SNAP labels, as well as introduced a chemical modification for better labeling of complex tissue. By combining these technologies, we were able to reveal that internalization strength and rate is ligand-dependent, and that GLP1R undergo heterogeneous ligand-induced internalization. Endogenous GLP1R were also found to form clustered and dynamic membrane nanodomains.

While SNAP-tags are widely used in cell biology applications, the efficacy of SNAP labels has largely been determined using cultured cells. However, we anecdotally noticed that a number of commonly used SNAP-tag labels performed poorly in GLP1R$^{SNAP/SNAP}$ compared to SNAP-GLP1R:AD293 cells (e.g. Alexa Fluor 647). As such, we systematically assessed several SNAP labels spanning different fluorophores, with distinct chemical and photophysical properties. We consistently observed that rhodamine derivatives performed best for SNAP labeling in tissue, with TMR and JF$_{549}$ producing the most specific signal. However, cytoplasmic GLP1R labeling was seen in GLP1R$^{SNAP/SNAP}$ islets above and beyond that expected under non-stimulated conditions, with some non-specific uptake also in GLP1R$^{WT/WT}$ islets. To increase the performance of the SNAP labels, we turned to a sulfonation strategy to render ligands cell impermeable[26,35], reasoning that more dye would be available to bind GLP1R at its primary cell surface location. For all dyes tested (TMR, JF$_{549}$, SiR and JF$_{646}$), installing a sulfonate moiety either on the benzylguanine or fluorophore itself led to vastly superior surface labeling. Even dyes such as Cy5, which in their BG-form appeared to primarily label cell organelles, were now able to exclusively label surface-exposed GLP1R. Together, these findings reveal differences between SNAP labeling in cultured cells and tissue, demonstrate the importance of SNAP label selection, and identify the best-performing labels for future studies with similar endogenously SNAP-tagged proteins.

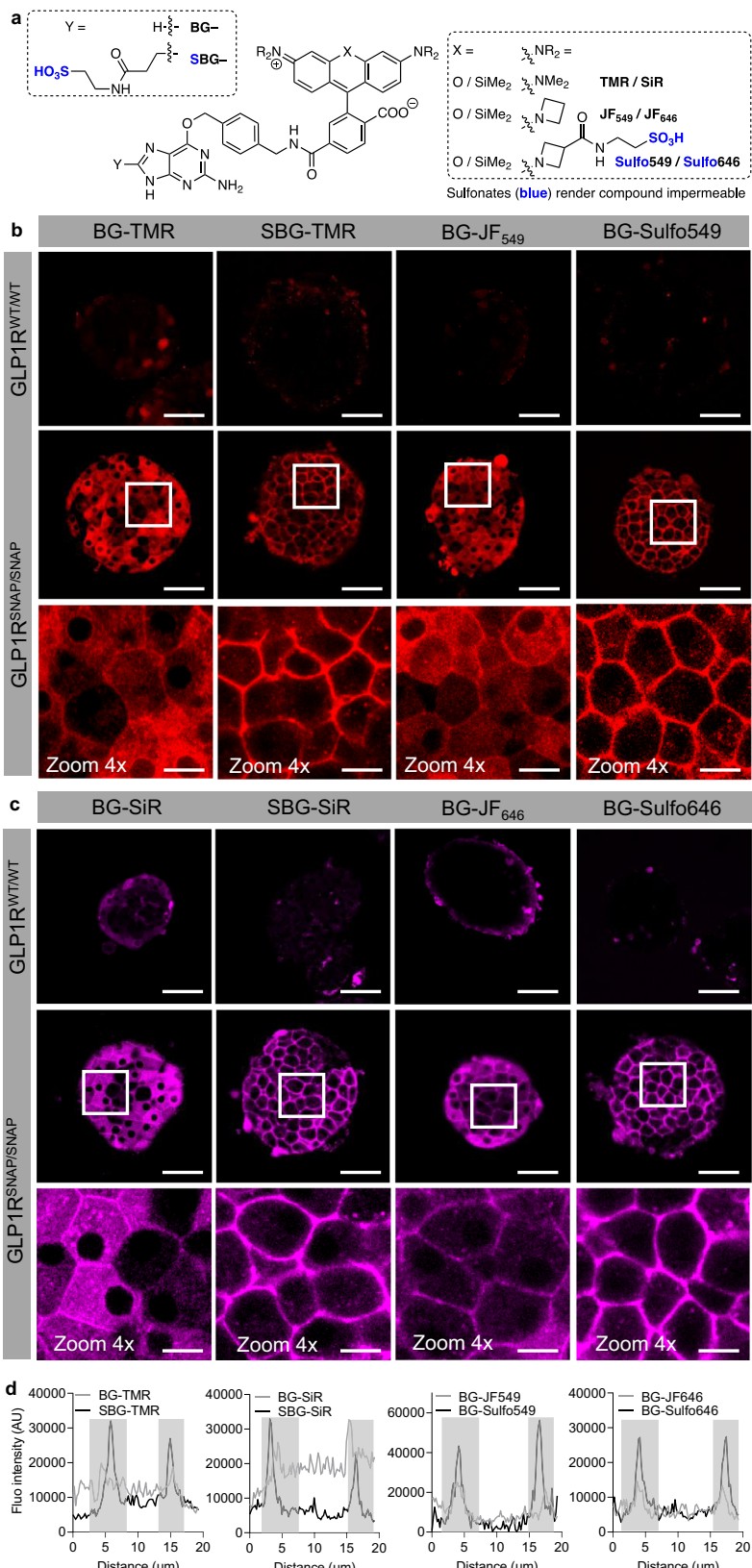

**Fig. 3 | Validation of cell impermeable rhodamine-bearing SNAP labels for visualization of surface protein endogenous targets in complex tissue.**
**a** Installation of a sulfonate, either on the benzylguanine or the dye itself, renders the SNAP label cell impermeable. **b** Compared to BG-TMR and BG-JF$_{549}$, SBG-TMR and BG-Sulfo549 lead to brighter and cleaner surface labeling of GLP1R$^{SNAP/SNAP}$ islets ($n$ = 12 islets, 3 animals for SBG-TMR; $n$ = 16 islets, 3 animals for BG-Sulfo549).

**c** Similar results are observed for SBG-SiR and BG-Sulfo646 versus BG-SiR and BG-JF$_{646}$ ($n$ = 13 islets, 5 animals for SBG-SiR; $n$ = 7 islets, 4 animals for BG-Sulfo646). **d** Representative line intensity profiles showing brighter, surface-localized labeling of GLP1R$^{SNAP/SNAP}$ islets with SBG- and Sulfo dyes (gray-shaded area shows membranous margins either side of the cell). Scale bar = 85 μm for all images, except zoom in scale bar = 17 μm. Source data are provided as a Source Data file.

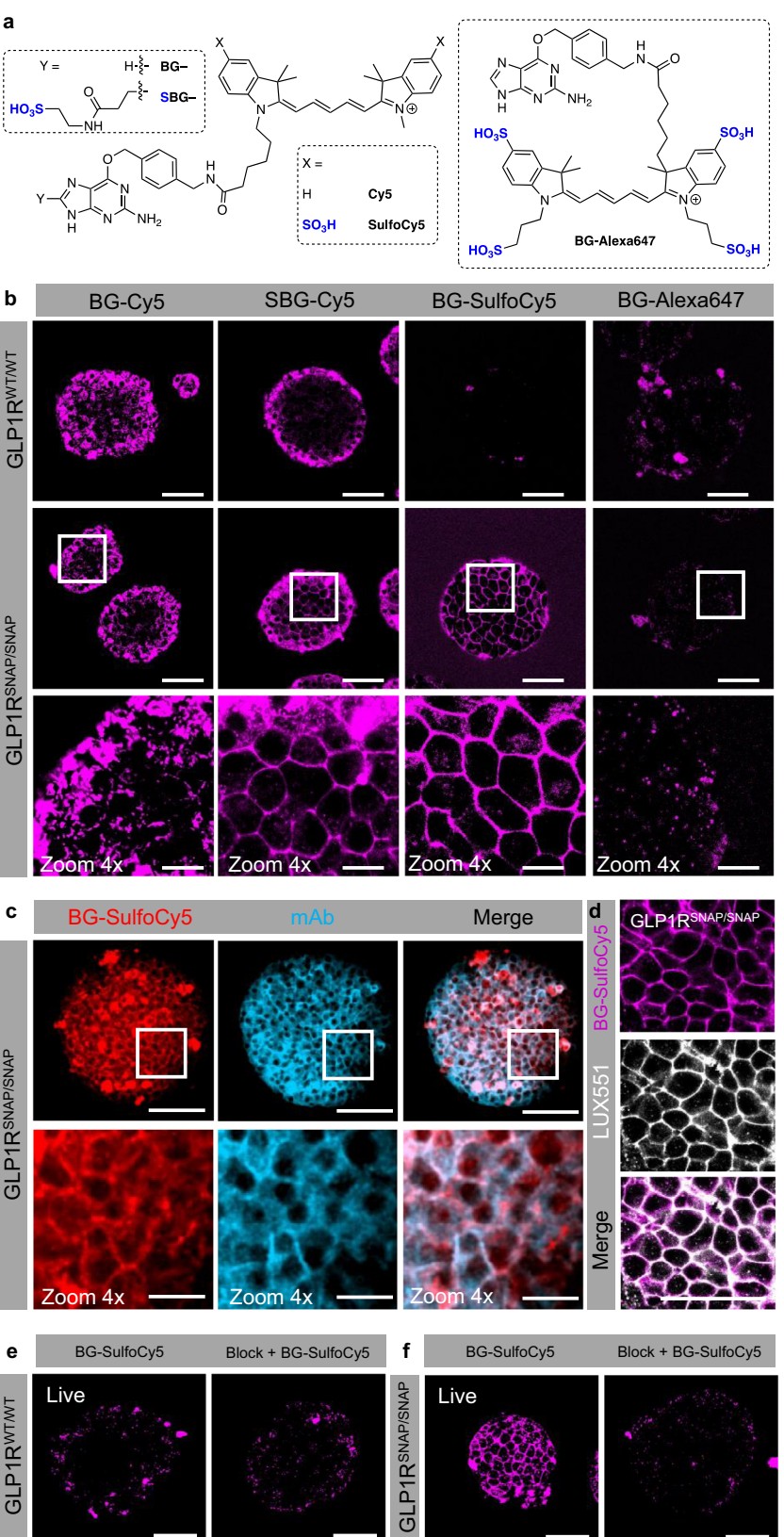

**Fig. 4 | Validation of cell impermeable cyanine-bearing SNAP labels for visualization of surface protein endogenous targets in complex tissue. a** Installation of a sulfonate, either on the benzylguanine or the dye itself, renders the SNAP label cell impermeable. **b** Sulfonation of BG-Cy5 allows the dye to label GLP1R$^{SNAP/SNAP}$ islets, with BG-SulfoCy5 showing the best performance compared to SBG-Cy5. The related cyanine BG-Alexa647, which is already sulfonated, shows no specific

labeling of GLP1R$^{SNAP/SNAP}$ islets ($n = 14$ islets, 4 animals). **c, d** BG-SulfoCy5 co-localizes with GLP1R mAb (**c**) and LUXendin551 (LUX551) (**d**) staining (GLP1R mAb, $n = 22$ islets, 10 animals; LUX551, $n = 13$ islets, 5 animals). **e, f** BG-Block has no effect on GLP1R$^{WT/WT}$ islets (**e**), but prevents BG-SulfoCy5 from labeling GLP1R$^{SNAP/SNAP}$ islets (**f**) ($n = 7$ islets, 3 animals). Scale bar = 85 μm for all images, except zoom in scale bar = 17 μm.

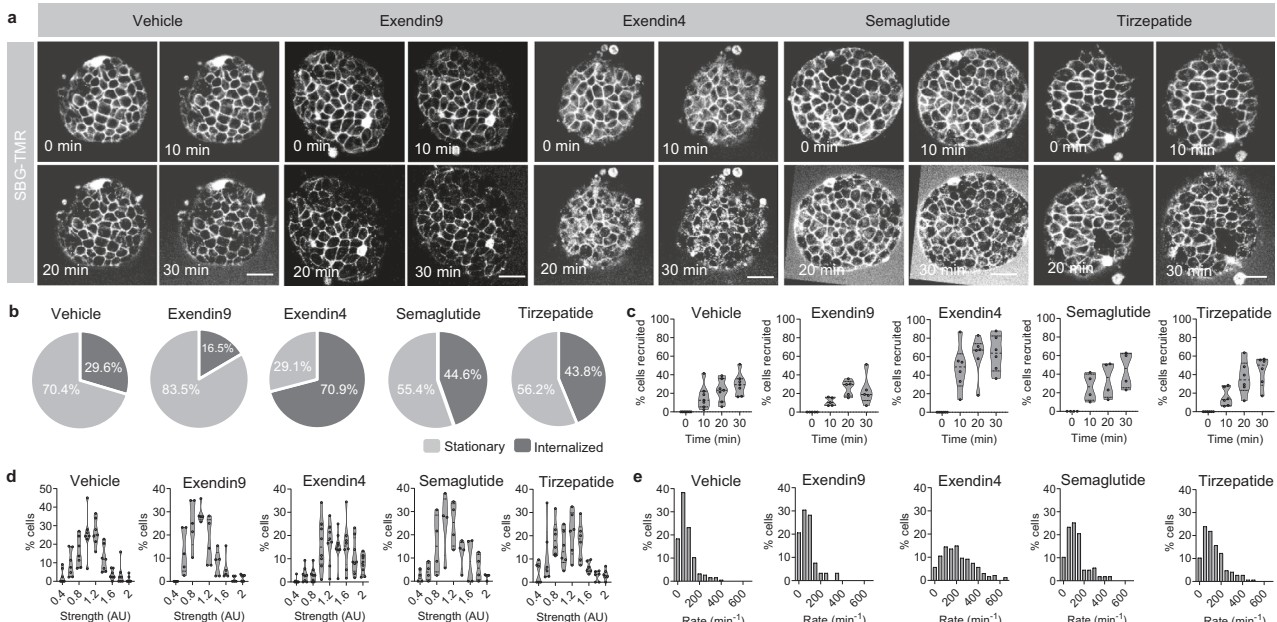

**Fig. 5 | Multicellular GLP1R dynamics are ligand-dependent. a** Representative images of SBG-TMR labeled islets at 0, 10, 20 and 30 min post-stimulation with vehicle, exendin9, exendin4, semaglutide or tirzepatide (scale bar = 34 μm). **b**, **c** Proportion of cells showing internalization (**b**) and recruitment of cells into internalization (**c**) in response to vehicle, exendin9, exendin4, semaglutide or tirzepatide after 30 min (*n* = 4 islets, 3 animals). **d**, **e** Frequency distribution of internalization strength (**d**) and internalization rate (**e**) across the imaged cell population following vehicle- or ligand-stimulation (*n* = 4 islets, 3 animals). Bar graphs show mean ± SEM. Violin plots show median and interquartile range. Source data are provided as a Source Data file.

For the studies here, we selected SNAP- over Halo-tag, since the SNAP_hGLP1R construct, used as the basis for the SNAP$_F$-mGLP1R template, is well characterized in terms of its pharmacology and trafficking in hundreds of studies. We note however that Halo-tags are advantageous in terms of rhodamine dye labeling and STED nanoscopy[35,42]. Ultimately, SNAP-tag is orthogonal to Halo-tag and using either protein self-label opens up the potential to simultaneously visualize two proteins (and more with CLIP-tag).

So far, most of our understanding about GPCR internalization/trafficking has necessarily been derived from experiments in heterologous cell systems, which allow SNAP/CLIP/Halo/GFP-tagged GPCRs to be dynamically visualized. However, cell lines do not re-capitulate the complex tissue environment nor allow truly endogenous GPCR to be studied. It is well accepted that tissues comprise heterogeneous cell states[3,43], even within cells derived from the same progenitor, as well as host intercellular communications[4,37]. Longitudinal imaging in intact pancreatic islets revealed a number of hitherto underappreciated features in terms of GLP1R trafficking/internalization. Firstly, 'constitutive' GLP1R trafficking/internalization was much higher than previously reported in immortalized cells[44]. This is likely due to the release of proglucagon-derived peptides by alpha cells, which are able to bind and activate the GLP1R in neighboring beta cells[45], and constitutive GLP1R cAMP signaling has been previously reported in beta cells[46]. Secondly, GLP1R internalization was found to be a heterogeneous cell trait, with subpopulations of beta cells being sequentially recruited into internalization by ligand. Similar findings have been shown for other cell signals such as Ca$^{2+}$ and cAMP[37,47,48], as well as insulin secretion itself[49,50], and likely reflect cell autonomous and non-autonomous heterogeneity. Thirdly, different ligands could be differentiated by the number of beta cells recruited into GLP1R internalization, GLP1R internalization strength and internalization rate, as well as the sequence of beta cell subpopulations engaged. Whereas Ex4 strongly recruited most of the beta cell population into GLP1R internalization, semaglutide and tirzepatide recruited only a subpopulation of beta cells, which responded earlier to input. In addition, beta cells recruited by semaglutide and tirzepatide displayed slow and weak GLP1R internalization. These findings fit with recent studies in cell lines showing reduced ligand-stimulated GLP1R internalization in response to semaglutide and tirzepatide[13,38]. Together, these studies show that pharmacological differences between ligands are preserved in primary tissue, whilst revealing at the same time a tertiary level of GPCR signaling at the multicellular level.

We previously showed that GLP1R possess higher organization at the nanoscopic level[17]. However, these data were obtained using MIN6 immortalized beta cells treated with antagonist, and whether similar organization is seen in primary islets has so far remained out of reach. Using *d*STORM nanoscopy, we reveal that endogenous GLP1R adopt a clustered formation at the beta cell membrane. While the functional relevance of clustering is unknown, we speculate that it might reflect signaling microdomains constrained by the cell cytoskeleton or membrane lipid domains[39,40,51]. Further suggesting the presence of signaling microdomains, single-molecule microscopy revealed that GLP1R at the membrane were highly mobile in their non-stimulated state, with a subpopulation becoming static or trapped following ligand stimulation. These results confirm previous findings in cell lines[17,39,52], and show that GLP1R higher organization is also a feature of endogenous receptors in complex tissue, as recently shown for the class C GPCR metabotropic glutamate receptor 4[53].

There are a number of limitations of the present study. Firstly, the comparison of dyes might only be relevant for detecting SNAP-GLP1R in the pancreatic islets and other tissues/GPCRs should be tested going forwards. Nonetheless, the analyses here provide interesting chemical insight into the dye properties likely to be useful in tissue. Secondly, while the mouse is a widely used pre-clinical model, it will be important to confirm results in human islets. To this end, the same limitations for GLP1R visualization apply, although islet re-aggregates have been shown to be amenable to CRISPR/Cas9, albeit for knock-out rather than knock-in of genes[54]. Thirdly, recent studies have shown the utility of DREADDs for the interrogation of endogenous FFA2[55]. While SNAP-tags are used predominantly for protein visualization, we envisage that GLP1R$^{SNAP/SNAP}$ tissue can in the future be combined with

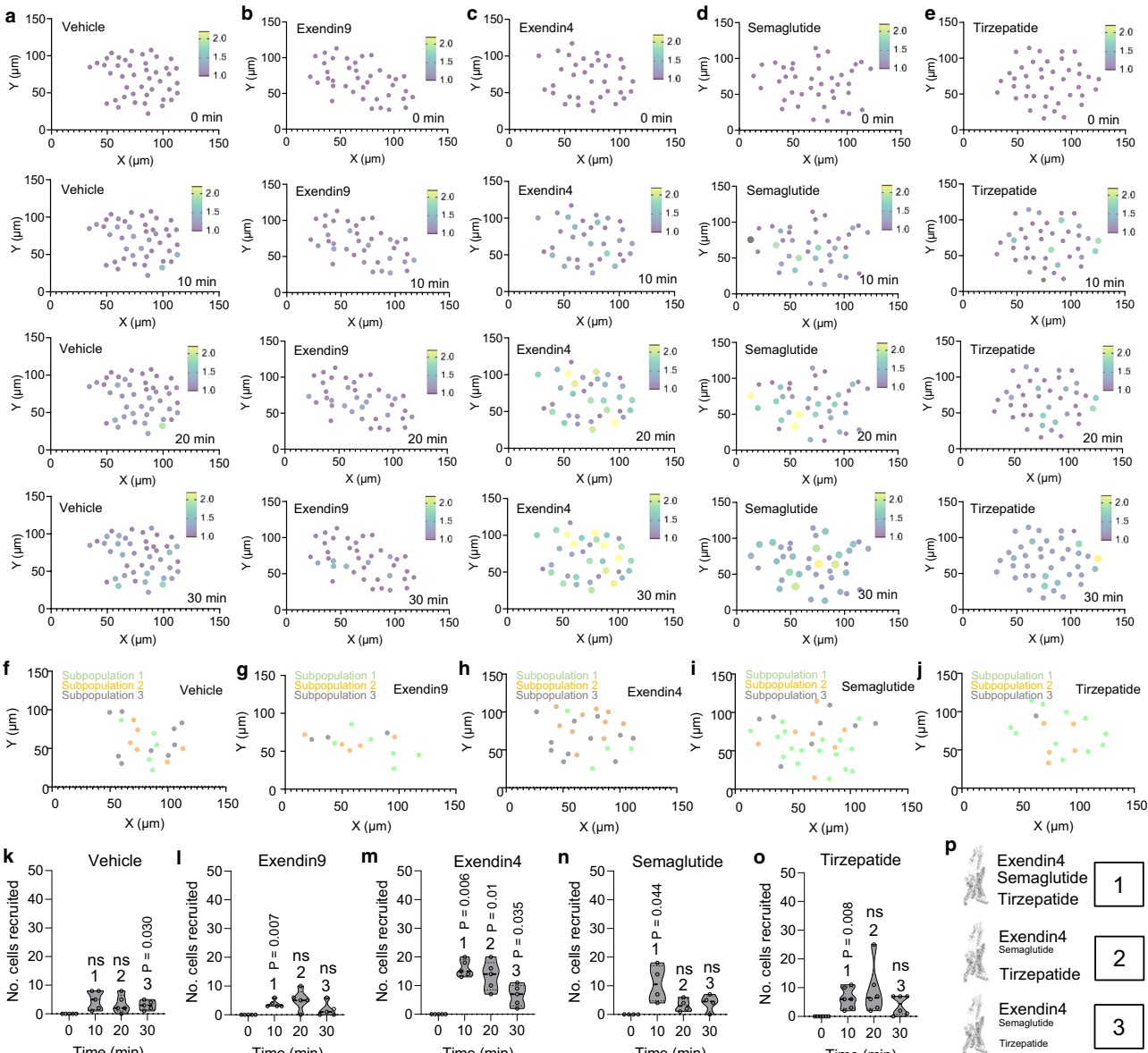

**Fig. 6 | GLP1R dynamics are heterogeneous. a–e** Cartesian (X-Y) maps for the islets in Fig. 5a showing cell position in the islet and changes in internalization strength over time following application of vehicle (**a**), or exendin9 (**b**), or stimulation with exendin4 (**c**), semaglutide (**d**) or tirzepatide (**e**) (representative plots from *n* = 4 islets, 3 animals). **f–j** Cartesian (X-Y) maps showing cell subpopulations recruited into internalization at 10 min (subpopulation 1), 20 min (subpopulation 2) and 30 min (subpopulation 3) post-stimulation with vehicle (**f**), exendin9 (**g**), exendin4 (**h**), semaglutide (**i**) or tirzepatide (**j**), for the islets shown in Fig. 5a and the graphs displayed in Fig. 5b–e. **k–o** Number of cells recruited into

internalization at each time point following stimulation with vehicle (**k**), exendin9 (**l**), exendin4 (**m**), semaglutide (**n**) or tirzepatide (**o**) (one-way RM ANOVA with Dunnett's or Bonferroni's post-hoc test; vehicle: *F* = 2.97, DF = 3; exendin9: F = 6.39, DF = 3; exendin4: F = 16.60, DF = 3; semaglutide: F = 6.99, DF = 3; tirzepatide: F = 3.16, DF = 3) (*n* = 4 islets, 3 animals). **p** Schematic showing relative effects of the various agonists on trafficking within cell sub-populations 1, 2, and 3 (font size = strength of effect). **P < 0.05, **P < 0.01, NS, non-significant. Violin plots show median and interquartile range. Source data are provided as a Source Data file.

(photoswitchable) tethered ligands for cell-specific interrogation of endogenous GPCRs[11,56]. Fourthly, trafficking was not correlated with second messenger generation or beta-arrestin recruitment at the individual cell level. Given the nature of the trafficking experiments performed here (3D multicellular timelapse) such multiparametric experiments are challenging and will require lightsheet imaging to achieve the frame rates required. Lastly, we focused our efforts on pancreatic islets as a tractable and relevant testbed for endogenous GPCR detection. However, GLP1RA and GLP1R/GIPR dual agonists, approved for the treatment of type 2 diabetes and more latterly obesity, have also shown efficacy for neurodegenerative and inflammatory disease states[7]. Yet localizing GLP1R protein and assigning cellular/molecular targets remains challenging due to lack

of specific and sensitive reagents[16]. We thus expect GLP1R^SNAP/SNAP mice and accompanying SNAP labels to inform tissue- and cell-specific GLP1R protein expression patterns.

In summary, we show the utility of self-labeling enzymes for the detection, visualization and interrogation of endogenous GLP1R at the protein level. We exploit this technology to update our knowledge about the tissue-level expression and regulation of GPCRs.

## Methods
### Ethical statement
Animal studies were regulated by the Animals (Scientific Procedures) Act 1986 of the U.K. (Personal Project Licences P2ABC3A83 and PP1778740). Approval was granted by the University of Birmingham's

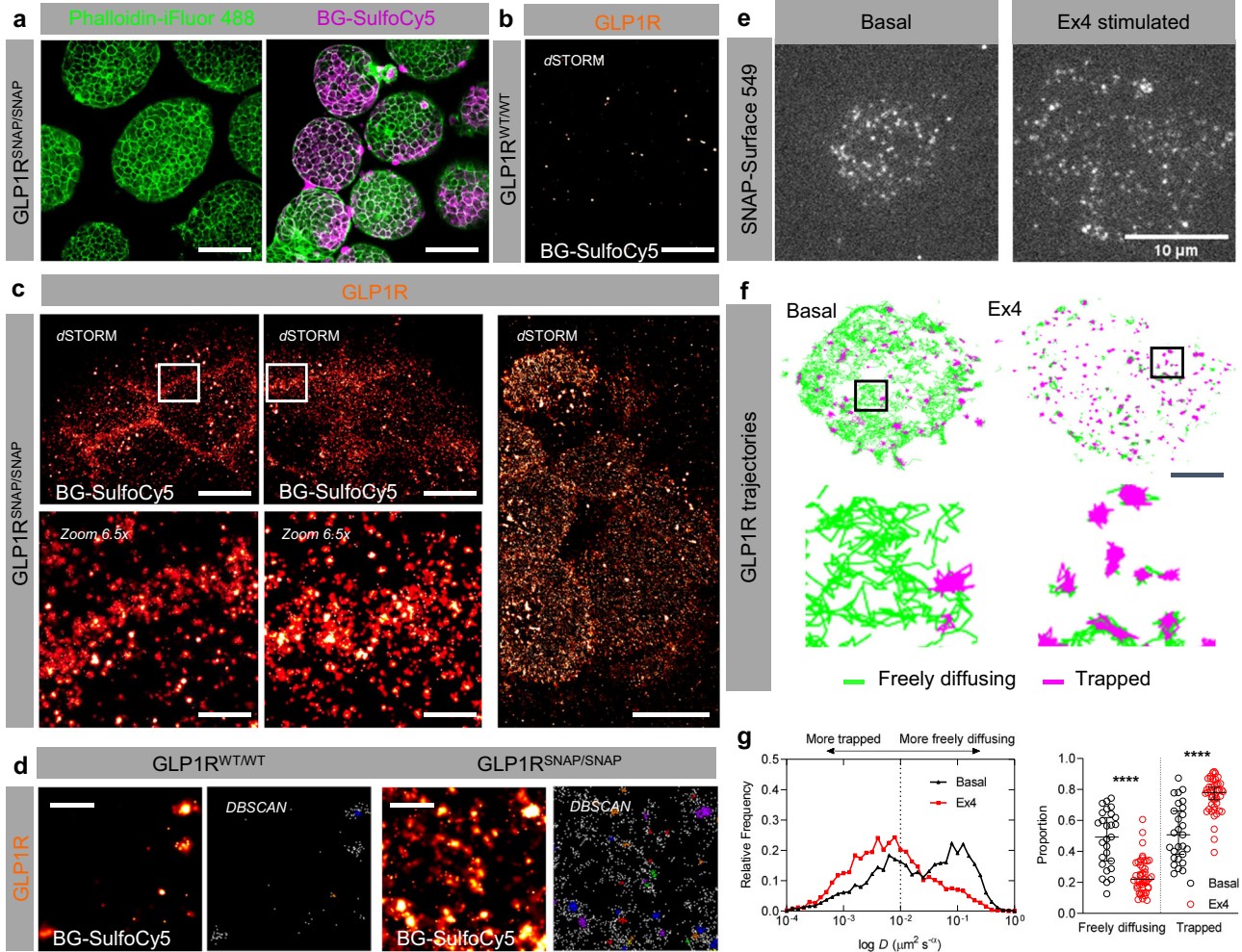

**Fig. 7 | GLP1R form nanodomains in primary tissue and cells. a** BG-SulfoCy5 signal is detected in GLP1R^SNAP/SNAP islets following formalin fixation and counterstaining for F-actin using Phalloidin-iFluor 488 (scale bar = 85 μm). **b** dSTORM signal is undetectable in GLP1R^WT/WT islets labeled with BG-SulfoCy5 (*n* = 4 islets, 3 animals) (scale bar = 85 μm). **c**, dSTORM reveals organization of GLP1R into nanodomains at the beta cell membrane in intact pancreatic islets (n = 8 islets, 3 animals) (scale bar = 8 μm, and scale bar = 1.2 μm for zoom). **d** DBSCAN analysis shows GLP1R clusters in GLP1R^SNAP/SNAP but not GLP1R^WT/WT islets (scale bar = 500 nm) (*n* = 4–8 islets, 3 animals) **e** Representative images from live-cell single molecule microscopy showing labeling of GLP1R in primary dissociated beta cells in non-stimulated and Ex4-stimulated states (*n* = 3 animals) (note that different cells were used for non-stimulated and stimulated conditions) (scale bar = 10 μm). **f** Representative maps showing GLP1R trajectories in non-stimulated and Ex4-stimulated states (green = freely diffusing; pink = trapped) (zoom-ins shown in the bottom panel, demarcated by a box; scale bar = 5 μm). **g** Measurement of diffusion coefficient (*D*) and trapping analysis reveal a shift from freely-diffusing to trapped receptors following Ex4-stimulation (two-sided Mann-Whitney test) (*n* = 27 trajectories, 3 animals). ****P < 0.0001. Scatter plot shows mean and 95% confidence interval. Source data are provided as a Source Data file.

Animal Welfare and Ethical Review Body (AWERB). All ethical guidelines were adhered to whilst carrying out this study.

## Chemicals
SNAP labels were previously reported[26,35,57,58]. SNAP labels are freely available for academic use subject to ability to supply.

## Generation and validation of GLP1R^SNAP/SNAP mice
A SNAP_f-tag was inserted into exon 1 of *Glp1r* using CRISPR-Cas9 genome-editing. The repair template was delivered as a single-stranded oligo DNA (ssODN) nucleotide in sense orientation. To generate the ssODN, SNAP_f was amplified from plasmid pET51b-SNAPfast[57] and cloned with flanking sequences homologous to *Glp1r* into pJet2.1 (Promega), followed by PCR amplification using primers 5′-CTCC TGCGCCTGGCGCTCCTGC-3′ and 5′-[Phos]AACACCGCAGCGTCCCCC TCC-3′. After digest with *Dpn*I, the double-stranded PCR product was subjected to Lambda exonuclease to remove the 5′-phosphorylated strand and generate a single-stranded repair template harboring SNAP_f flanked by 55 nt homology arms on either side. The 680 bases repair template does not contain the full guide RNA targeting sequence and no PAM sequence. A modified single-guide RNA (sgRNA, synthego) with the guide sequence AGGGCCGGCCCCCGCCCUC targets *Glp1r* for Cas9 to cut 3-4 nucleotides downstream of the insertion site. Both sgRNA (20 ng/μL) and ssODN (30 ng/uL) in microinjection buffer (10 mM Tris-HCl pH 7.5, 0.1 mM EDTA in HPLC-pure water) were injected into the pronucleus of 1-cell embryos from super-ovulated Cas9-overexpressing mice (strain Gt(ROSA)26Sortm1.1(CAG-cas9*,–EGFP)Fezh/J; JAX stock no. 024858). Following culture overnight, ~80% of embryos reached the 2-cell stage and were implanted into CD1 surrogate female mice. Genotyping was used to confirm mice that integrated repair template (SNAP-GLP1R) followed by PCR amplification of the on-target site and Sanger sequencing: six pups were born and we could detect three deletion (1–27 bp) and three insertion events. Only one inserted SNAP_f-tag was without any mutation while the other two contained either a point mutation leading to an amino acid change or an additional inversed sequence from exon1 within the tag. One founder was taken forward and backcrossed to C57BL6/J. The CRISPR Guide Design Tool (crispr.mit.edu) was used to

predict potential off-target sites. The seven top hits (2-4 mismatches to the guide sequence) and further three lying within genes (3-4 mismatches) were amplified by PCR followed by Sanger sequencing (Supplementary Table 1 and Fig. 1). The founder mouse was bred for 1–2 generations to outbreed the Rosa26-Cas9 allele as well as the affected off-target. Genotyping was performed using Sanger sequencing or PCR. Animals were bred as heterozygous pairs to ensure GLP1R[WT/WT] littermates. Mice were socially-housed in specific-pathogen free conditions under a 12 h light-dark cycle with ad libitum access to food and water, relative humidity $55 \pm 10\%$ and temperature $21 \pm 2\,°C$.

## Oral glucose tolerance testing

Mice aged 8 weeks were fasted 4–6 h, before receiving 2 g/kg oral glucose using a flexible gavage tube. Blood samples were taken from the tail vein at 0, 15, 30, 60, 90, and 120 min post glucose-challenge. Glucose concentrations were measured using a Contour XT glucometer (Ascensia Diabetes Care, Switzerland).

## cAMP assays

Potency for cAMP generation and inhibition was tested in AD293 cells (Agilent, Cat# 240085) transiently transfected with YFP and either SNAP-GLP1R (Cisbio)[17] or GLP1R-GFP[59]. The cell line is not listed on the International Cell Line Authentication Committee database. Briefly, cells were incubated with increasing concentrations of ligand for 30 min + 100 $\mu$M IBMX, before lysis and measurement of cAMP using LANCE cAMP assays (PerkinElmer) according to the manufacturer's instructions. TR-FRET signal was detected at 665 nm (excitation 340 nm) using a BMG LABTECH PHERAstar microplate reader. $EC_{50}$ values were calculated using log concentration–response curves fitted with a three-parameter or four-parameter equation.

## Islet isolation

Mice aged 7–12 weeks were humanely euthanized by cervical dislocation, before bile duct injection of collagenase solution in RPMI (1 mg/ml; Serva NB8). Inflated pancreata were removed, digested for 10–12 min at 37 °C in a water bath, before separation of islets using a Ficoll or Histopaque gradient. Islets were cultured (5% $CO_2$, 37 °C) in growth medium (RPMI supplemented with 10% FCS, 100 units/mL penicillin, and 100 $\mu$g/mL streptomycin).

## Insulin secretion measures

Batches of ten mouse islets were placed in low protein-bind 1.5 ml Eppendorf tubes containing 0.5 ml HEPES-bicarbonate buffer supplemented with 3 mM glucose and 0.1% BSA and allowed to acclimate for 1 h. Buffer was then removed before sequential addition of 3 mM glucose, 16.7 mM glucose or 16.7 mM glucose + 20 nM Ex4 (AnaSpec Cat# ANA24463), and incubation for 30 min at 37 °C (i.e. paired measures). Insulin concentrations were measured using an ultra-sensitive HTRF assay kit (PerkinElmer Cat# 62IN2PEG).

## cAMP imaging

Islets were transduced with a relatively pH-insensitive Epac2-camps Cerulean/Citrine FRET probe (a kind gift from Prof. Dermot Cooper, Cambridge)[60]. After 2 days, islets were imaged in HEPES-bicarbonate buffer containing 17 mM glucose and stimulated with 20 nM Ex4 (biotechne, Cat# 1933). cAMP imaging was performed using a CrestOptics X-Light spinning disk and 10×/0.4 numerical aperture (NA) objective mounted on a Nikon Ti-E automated microscope base. Excitation was delivered at 445 nm using a North 89 LDI, with emitted signals detected using a Photometrics Delta Evolve EM-CCD at $\lambda = 460-500$ and $\lambda = 520-550$ nm for Cerulean and Citrine, respectively. Epac2-camps intensity was calculated as the ratio of Cerulean/Citrine. Traces were presented as $R/R_0$ where $R$ = ratio at any timepoint and $R_0$ = average ratio for the first 20 frames.

## SNAP labeling

GLP1R[SNAP/SNAP] islets were incubated with 500 nM SNAP label for 30 min at 37 °C and 5% $CO_2$ in growth medium, before washing three times. When BG-Block was used, islets were incubated in growth medium supplemented with 1 $\mu$M BG-Block for 30 min at 37 °C and 5%, before addition of SNAP label. Islets from GLP1R[WT/WT] littermates were used as controls. Imaging was performed using Zeiss LSM780 or LSM880 meta-confocal microscopes equipped with a 40x water NA 1.2 Korr FCS M27 objective, GaAsP spectral detectors and an environmental chamber (37 °C, 5% $CO_2$). Pixel size was 0.21 $\mu$m x 0.21 $\mu$m over a $1024 \times 1024$ field of view, line average x 2. Excitation / detection wavelengths were as follows for the various tested dyes: BG-OG – 488 nm / 493–598 nm; (S)BG-TMR – 561 nm / 566–685 nm; BG-JF$_{549}$ and BG-Sulfo549 – 561 nm / 569-649 nm; BG-CPY – 594 nm / 625–740 nm; (S)BG-SiR – 633 nm / 641-694 nm; BG-JF$_{646}$ and BG-Sulfo646 – 633 nm / 641-694 nm; BG-SulfoCy5 – 633 nm / 641-694 nm; BG-Alexa647 – 633 nm / 641-694 nm. SNAP labeling was analyzed on raw images using a mean intensity along a line profile. If required, Gaussian denoising was performed. Islets labeled with BG-Block/BG-Sulfo549 were imaged using a CrestOptics X-Light spinning disk equipped with a 20x/0.75 NA objective mounted on a Nikon Ti-E automated microscope base. Excitation was delivered at $\lambda = 543-558$ nm using a Lumencor Spectra X, with emitted signals detected at $\lambda = 570-640$ nm.

## LUXendin labeling and immunostaining

GLP1R[SNAP/SNAP] islets were incubated for 30 min with 100 nM LUXendin551/LUXendin645 and 500 nM BG-TMR/BG-JF$_{549}$/BG-SulfoCy5 in culture medium. For immunostaining, islets were fixed in 4% formaldehyde fixation for 12–15 min before application of mouse monoclonal anti-GLP1R antibody (Iowa DHSB; mAb #7F38, 1:30) overnight at 4 °C, diluted in PBS + 0.1% Triton + 1% BSA. Goat anti-mouse DyLight488 secondary antibody (ThermoFisher Scientific; Cat #35502, 1:1000) was applied for 1 h at room temperature followed by mounting with Vectashield Hardset containing DAPI. Imaging was performed identically to above, with excitation and detection wavelengths for LUXendin645 – 633 nm / 641-694 nm; DyLight488 – 488 nm / 490–561 nm; DAPI – 405 nm / 410-498 nm. If required, Gaussian denoising was performed. LUXendins are available from Celtarys Research.

## GLP1R internalization studies

GLP1R[SNAP/SNAP] islets were incubated with 500 nM SBG-TMR for 30 min at 37 °C and 5% $CO_2$ in growth medium, before washing three times. Islets were imaged using a Zeiss LSM780 meta-confocal microscope, as above, but with a 63x water NA 1.2 Korr M27 objective. Pixel size was 0.13 $\mu$m x 0.13 $\mu$m over a $1024 \times 1024$ field of view, line average x 2. Timelapse z-stacks (50 $\mu$m) were analyzed using ImageJ and Fiji (NIH). Histograms were equalized to account for photobleaching and denoising performed using a median filter with radius 1 pixel and threshold 50. Movement was corrected using the StackReg plugin and a rigid body transformation[61]. Regions of interest (ROI) were then placed over the cytoplasm of GLP1R + (i.e. SNAP-labeled) cells before extraction of the mean gray value (MGV) at 0, 10, 20 and 30 min. Since labeling intensity and cell size were similar for all states examined, MGV quantitatively reports cytoplasmic SNAP labeling i.e., that derived from internalized GLP1R. To report changes in intensity over time, MGV was normalized for each individual cell versus baseline (i.e. 0 min). To identify the proportion of cells recruited into internalization, a 20% threshold was used. To depict the time course of internalization within each individual cell, a color map was used spanning zero (1.0), low (1.2–1.5) medium (1.5–2.0) and high (>2.0) levels of internalization. Semaglutide (NNC0113-0217) was provided by Novo Nordisk Compound Sharing. Tirzepatide was a kind gift from Dr Ben J. Jones (Imperial College London).

## *d*STORM imaging

Pancreatic islets from GLP1R[SNAP/SNAP] and GLP1R[WT/WT] mice were labeled in growth medium supplemented with 500 nM BG-SulfoCy5 at 37 °C and 5% $CO_2$ for 1 h. After three washes in PBS islets were fixed in 4% PFA at room temperature for 12-15 min, followed by three washes in PBS and blocking in PBS containing 0.1% Triton-X-100 and 1% BSA at room temperature for 1 h. In the same buffer, islets were stained for actin using 1:200 Phalloidin-iFluor 488 (ab176753, abcam) followed by two washes in PBS. Islets were embedded in a μ-slide 8-well chamber slide (ibidi) using 1% agarose in water.

*d*STORM imaging was performed on an Oxford Nanoimaging (ONI) Nanoimager. The sample was first immersed in *d*STORM oxygen scavenging buffer system (10 mM Tris with 50 mM NaCl (pH 8.0) supplemented with 10% (w/v) glucose, 168.8 AU/ml of glucose oxidase (G2133; Sigma), 1404 AU/ml of Catalase (C100; Sigma), and 10 mM cysteamine (30070; Sigma)). The sample was initially pumped with 650 nm laser excitation at full power (120 mW) using epifluorescent illumination to pump SulfoCy5 to the dark state, and then imaged in HILO illumination at 65 mW. SulfoCy5 either transitioned back from the dark state naturally, or low power 405 nm excitation (from 0.1 to 0.5 mW) was used to elicit the transition. The emission was collected on one half of Hamamatsu sCMOS camera chip using a LP650 filter. Images (256 × 256 pixels, pixel size = 117 nm) were collected with 50 ms frame integration time, for between 3000 and 10000 frames (acquisitions were ended upon cessation of dye blinking). Single molecule localization of SulfoCy5 emissions were carried out in Fiji using the ThunderSTORM plugin. Clustering of localizations was analyzed using DBSCAN in MATLAB. DBSCAN parameters were: ε = 25 nm (the average precision of the data), and minPts = 8.

## Single molecule tracking

GLP1R[SNAP/SNAP] islets were dissociated using trypsin-EDTA (0.25%) and then plated onto 25 mm clean glass coverslips coated with poly-L-lysine. The next day cells were incubated with SNAP-Surface549 (New England Biolabs) in complete culture medium for 20 min at 37 °C and washed 3 × 5 min using HEPES-bicarbonate buffer, leading to labeling of ~80% of surface GLP1R required for single-molecule analysis. GLP1R[SNAP/SNAP] islet cells were then imaged in HEPES-bicarbonate buffer supplemented with 16.7 mM D-glucose using a custom built TIRF microscope (Cairn Research) comprising an Eclipse Ti2 (Nikon, Japan) base with EMCCD camera (iXon Ultra, Andor), 561 nm diode laser, and a 100x oil-immersion objective (NA 1.49, Nikon). Image sequences were acquired with an exposure time of 30 ms. Since laser exposure leads to non-linear photobleaching, basal and stimulated conditions were studied in separate cells to allow comparison of GLP1R trajectories over the same time course and laser exposure. Automated single-particle detection and tracking were performed with the u-track software[62] and the obtained trajectories were further analysed using custom algorithms in the MATLAB environment, as previously described[40]. Sub-trajectory analysis of trapped and free portions was performed using a method based on recurrence matrix[63].

## Software

Images were captured using Metamorph 7.7 (Molecular Devices), Zen 2012 (Zeiss) and NimOS 1.19.3 (ONI). Image analysis was performed using ImageJ 1.5j8 (NIH), Zen 3.5 (Blue Edition; Zeiss) and MATLAB 2017b. Numerical data were analyzed using MATLAB R2017b (Mathworks), R Studio (R Project) and Prism 9 (Graphpad).

## Statistics and reproducibility

To exclude any sex-specific effects of GLP1R[SNAP/SNAP] knock-in, male and female animals were considered separately for metabolic phenotyping experiments. All in vitro studies combined tissue from both sexes, since: 1) no sex-specific phenotype was observed in vivo; and 2) male and female tissues were indistinguishable by their SNAP-tag labeling or GLP1R staining.

Measurements were performed on discrete samples unless otherwise stated, with data normality assessed using D'Agostino-Person test. Pairwise comparisons were made using Student's two-tailed unpaired or paired t-test. Multiple interactions were determined using one-way ANOVA or two-way ANOVA, adjusted for repeated measures where relevant. Pairwise post-hoc testing was performed using Bonferroni's or Dunnett's test. Degrees of freedom were accounted for during all post-hoc testing. A p-value less than 0.05 was considered significant.

The independent replicate is isolation of islets from a single animal. In all cases, experiments use islets from at least three animals, taken from at least three separate isolation procedures, which is a nuisance variable and hence used as a blocking factor. Since variation between individual islets/cells is greater than variation of the mean between animals, they are considered as separate datapoints for analysis purposes. For representative images, the experiment was repeated the same number of times as the related quantification, always with similar results.

Where the n number occupies a range of samples or animals, the lowest value is provided as per journal guidelines. Hence, the number of datapoints on the graph may be higher than the stated sample size.

## Reporting summary

Further information on research design is available in the Nature Portfolio Reporting Summary linked to this article.

## Data availability

All data extracted from raw images is provided in the source data file. Due to their large size, individual raw image files are available upon request from J.B. or D.J.H., who will respond within 30 working days. GLP1R[SNAP/SNAP] mice are subject to a Material Transfer Agreement. Source data are provided with this paper.

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

## Acknowledgements

D.J.H. was supported by MRC (MR/N00275X/1 and MR/S025618/1) and Diabetes UK (17/0005681) Project Grants, as well as a UKRI ERC Frontier Research Guarantee Grant (EP/X026833/1). This project has received funding from the European Research Council (ERC) under the European Union's Horizon 2020 research and innovation programme (Starting Grant 715884 to D.J.H.) and under the European Union's Horizon Europe Framework Programme (deuterON, Grant agreement No. 101042046 to J.B.). The research was funded by the National Institute for Health Research (NIHR) Oxford Biomedical Research Centre (BRC). Work in the D.C. lab was supported by a Wellcome Trust Senior Research Fellowship (212313/Z/18/Z to D.C.). The views expressed are those of the author(s) and not necessarily those of the NHS, the NIHR or the Department of Health. We thank the Microscopy Facility at Birmingham University for support and maintenance of microscopes, Biomedical Services Unit Birmingham University for assistance with production of genetically-altered mouse models, as well as Bettina Mathes and Kai Johnsson (both MPIMR) for providing reagents. The funders had no role in paper design, data collection, data analysis, interpretation or writing of the paper.

## Author contributions

J.A., D.N., N.H.F.F., D.J.N., Z.K., Y.L., F.C., K.V., and N.L. performed experiments and analyzed data. J.A. generated the SNAP_mGLP1R mice. A.B. provided advice on construct design and performed micro-injections. P.N.N., D.C., D.O., D.J.H., and J.B. supervised the studies. J.A., J.B., and D.J.H. conceived and designed the studies. J.A., J.B., and D.J.H wrote the paper with input from all authors.

## Competing interests

J.B. and D.J.H. receive licensing revenue from Celtarys Research for provision of chemical probes. J.B. is a consultant for Vedere Bio II, Inc. The remaining authors declare no competing interests.
