## [Peer Review File · Nature Communications]

Revealing the tissue-level complexity of endogenous glucagon-like peptide-1 receptor expression and signalingREVIEWER COMMENTS

Reviewer #1 (Remarks to the Author):

Ast and colleagues describe the CRISPR Cas9 mediated generation of a N terminally tagged GLP-1 receptor for monitoring and tracking of endogenously expressed receptors. As the authors rightly point out, this is a bottleneck in the GPCR discovery process, with most information on higher order structures/ trafficking/signalling obtained from heterologous cell systems over-expressing GPCRs. The utilisation of this approach is innovative and will aid taking important next steps in understanding the endogenous regulation of this T2DM drug target. Moreover, this provides important insights that are applicable to the wider GPCR superfamily of receptors, which are highly important drug targets.

For study specifics, the SNAP-GLP1R mouse model was characterised using physiological parameters appropriate for the GLP1R, with no differences observed between wild type and SNAP-GLP1R mice. In vitro expression of the SNAP-GLP1R in AD293 cells showed no difference in cAMP production in comparison to the WT construct in terms of cAMP production. These data presented appear robust. However, there are some controls that would be prudent to include. A basic work-up of the endogenous G protein-dependent signaling in islets of the SNAP-GLP1 mice is missing (likewise G protein-independent) and would add robustness and reassurance that endogenous SNAP-GLP1R signalling is intact.

The utility of different fluorophores for imaging endogenous GLP1R was examined with interesting non-specific uptake/lack of specificity of many of the dyes tested, including the commonly used dSTORM dye Alexa 647, providing useful information for tissue-labelling versus cell monolayers.

Next, the authors investigated ligand-specific spatial-temporal differences in GLP1R trafficking in islets and aspects of GLP1R dynamics and nanodomain organisation. Their results present some interesting concepts, showing constitutive GLP1R activation. Moreover, the heterogeneous trafficking responses of the GLP1R within islets to distinct ligands. At a nanoscopic level, distinct clustering of the GLP1R receptor was also observed. These raise some interesting concepts supporting individual cellular regulation of responses within intact tissues, with potential for discrete signalling and physiologic consequences.

An important question arises, given the distinct cellular spatial-temporal regulation observed in islets of both GLP1R trafficking, can these be correlated with differences observed in second messenger generation and b-arrestin recruitment at a cellular level?

The authors hint at differential tissues in their discussion. Given the controversies in tissue-specific expression of GLP1R, can the authors comment on the tissue distribution of the receptor in this mouse model?

Overall, the methodologies utilised are cutting edge and appropriate for the research questions and study well controlled and carried out.

Reviewer #2 (Remarks to the Author):

Ast et al. created a mouse with a SNAP-tag knocked into the N-terminus of the GLP1 receptor. The authors tested different dyes in pancreatic islets, an example for primary tissue staining. Different tests and stimulations show that the knock-in is functional.

The testing is basically well done, but there are several points where details are missing or the explanation should be expanded. To be able to classify the quality and specificity of the labelling I had difficulties to see single cells in the small images. The images could be easily expanded showing magnifications of single cells so that the amount of cytosolic to membrane labeling could be estimated better. This could be shown in the supplement. It is also not clear to me why there is such a large amount of cytosolic label; for example, in Fig. 2b, d and e, the labeling with BG-TMR seems to be strong in the cytosol and not just in the membrane. I guess the dark spots are the nuclei. Why is this so strong? Is this specific labeling of internal receptor or unspecific background? Would more washing steps help? Is this as strong in single, dissociated cells or only in the tissue?

List of comments:

- Halo-tags might be better than SNAP-tags for far-red rhodamines as shown in

<https://doi.org/10.1016/j.chembiol.2019.01.003> This should be at least discussed or tested in cell

culture.

- Is the labeling the same throughout the whole islet? To show that the substrate penetrates the whole islet a profile through the whole islet should be shown or several images at different depth in the supplement.
- The comparison with LUX645 (Fig. 2e) should be shown on much higher magnification (single cells) to see if it labels the same structure.
- To me it is not clear what the antibody labels in Fig. 2e., it seems to be everywhere. As pointed out above, this needs to be magnified. Both images, BG-JF549 and mAb are in mixed colors, this is difficult. At least a color map is needed.
- The magnifications in Fig. 2b, 3b, 3e and others should show a scale bar.
- Figure 3 legend: "SBG-TMR and BG... lead to brighter and cleaner labeling...". This should be shown and explained. The brighter labeling should be shown, for example, with a scale bar of the fluorescence signal. I would not expect that the labeling of receptors in the membrane is different between both substrates (with and without the sulfonate). If it is different, is this due to the labeling efficacy or the quantum yield of the fluorophore? However, the latter should not change by modifying the BG linker.
- "Cleaner signal" such as mentioned in line 135 or "cleaner labeling" in line 680 is not a good scientific term and should be better explained /defined.
- On the same line, "... more specific than ..." in line 136 should be better explained.
- Restricting the labelling to the surface is in principle good, but it needs to be clear that it is specific. Why not performing dSTORM also with the LUXendins? Or the mAbs?
- DBSCAN should be explained in 1-2 sentences.
- Fig. 6c: The colour map is red-white, DAPI in white would be difficult. I am not sure if there is a DAPI labeling in these images? Scale bar should be added to the zoom images.
- Fig. 6e: Are both images showing the same cell?
- Line 163-165: Verb missing.
- Line 391: "Detection" instead of "emission". The fluorescence emission is broader.

Reviewer #3 (Remarks to the Author):

The manuscript of Ast and colleagues reports a new mouse model in which GLP-1 receptor expression and signaling can be monitored. The authors use a SNAP-tagging approach and report validation studies as well as an initial pass of characterization of the expression and tracking of GLP-1 receptors in tissues including pancreatic islets. Complex responses are observed, both with respect to heterogenous cell types, as well as specific identified GLP-1R agonists. The studies are well thought out, and appropriate controls are utilized and presented.

I offer the following suggestions for improvement:

1. The results section is written quite superficially throughout. Statistical findings are not presented beyond occasional statement of whether something was statistically significant or not. Complex panels of graphs are only vaguely described. As just one example, Figure 1b and 1c present crucial validation studies but are described in the text only as "The construct was initially tested in vitro and found to signal and localize identically to human GLP1R-GFP and SNAP-GLP1R constructs widely used in cell biology applications (Figure 1b and c)." To be clear, this is just a single example; the results section should be thoroughly re-written to describe the presented data in much greater detail. Adding specific statistical results into figure legends similar to what the group presented in Ast et al., Nat Commun. 2020 Jan 24;11(1):467. doi: 10.1038/s41467-020-14309-w would also be very helpful in this regard.

2. Statistical tests are defined well in Methods, however it is unclear in what cases replicates are based in something other than an individual animal. In some cases it appears that replicates were treated as something else, for example an individual pancreatic islet and/or culture preparation rather than an animal. Please make clear, rigorous and transparent and ensure that in vivo/ex vivo data are based on a mouse = n of 1.

3. Recommend moving the Exendin9 data from Suppl Figure (esp panels Suppl Figure 2a and 2d) into main Figure 4. These data are very interesting re: constitutive activity and would be best presented and interpreted with direct comparison to the agonists.

We thank the three expert Reviewers for providing constructive feedback on our studies. We also thank the Editor for their efficient handling of the manuscript. We have responded in full to the Reviewers' comments by providing further data, analysis and discussion where appropriate. With these revisions, we hope that the manuscript will now be acceptable for publication in *Nature Communications*. A point-by-point response is provided below.

Reviewer #1 (Remarks to the Author):

Ast and colleagues describe the CRISPR Cas9 mediated generation of a N terminally tagged GLP-1 receptor for monitoring and tracking of endogenously expressed receptors. As the authors rightly point out, this is a bottleneck in the GPCR discovery process, with most information on higher order structures/ trafficking/signalling obtained from heterologous cell systems over-expressing GPCRs. The utilisation of this approach is innovative and will aid taking important next steps in understanding the endogenous regulation of this T2DM drug target. Moreover, this provides important insights that are applicable to the wider GPCR superfamily of receptors, which are highly important drug targets.

We thank the reviewer for their constructive and kind comments. Along these lines, we have begun to process the paperwork for deposition of GLP1R^{SNAP/SNAP} mice into the European Mouse Mutant Archive, and have already made available the SNAP labels to numerous labs.

For study specifics, the SNAP-GLP1R mouse model was characterised using physiological parameters appropriate for the GLP1R, with no differences observed between wild type and SNAP-GLP1R mice. In vitro expression of the SNAP-GLP1R in AD293 cells showed no difference in cAMP production in comparison to the WT construct in terms of cAMP production. These data presented appear robust. However, there are some controls that would be prudent to include. A basic work-up of the endogenous G protein-dependent signaling in islets of the SNAP-GLP1 mice is missing (likewise G protein-independent) and would add robustness and reassurance that endogenous SNAP-GLP1R signalling is intact.

The reviewer makes a good point. While we showed that Ex4-stimulated insulin secretion was intact in GLP1R^{SNAP/SNAP} mice, we did not look at upstream secretion amplifying pathways mediated primarily by cAMP. We now show that the GLP1R agonist Ex4 stimulates similar magnitude cAMP rises in GLP1R^{WT/WT} and GLP1R^{SNAP/SNAP} islets incubated at 17 mM glucose concentration (Figure 1m-p). These data show that introduction of the N-terminal SNAP-tag is unlikely to influence GLP1R signaling and further validate the model.

The utility of different fluorophores for imaging endogenous GLP1R was examined with interesting non-specific uptake/lack of specificity of many of the dyes tested, including the commonly used dSTORM dye Alexa 647, providing useful information for tissue-labelling versus cell monolayers.

Next, the authors investigated ligand-specific spatial-temporal differences in GLP1R trafficking in islets and aspects of GLP1R dynamics and nanodomain organisation. Their results present some interesting concepts, showing constitutive GLP1R activation. Moreover, the heterogenous trafficking responses of the GLP1R within islets to distinct ligands. At a nanoscopic level, distinct clustering of the GLP1R receptor was

also observed. These raise some interesting concepts supporting individual cellular regulation of responses within intact tissues, with potential for discrete signalling and physiologic consequences.

An important question arises, given the distinct cellular spatial-temporal regulation observed in islets of both GLP1R trafficking, can these be correlated with differences observed in second messenger generation and b-arrestin recruitment at a cellular level?

The reviewer makes a good point. We note however that the trafficking experiments shown here provide the first description of GLP1R internalization/fate in the same cell, across hundreds of cells within the live tissue, already a very high bar. A major challenge of these studies is that z-stacks (~ 50 um) are needed at each timepoint to accurately assign trafficking to individual cells as well as adjust for focus drift. As such, simultaneous measurement of second messenger generation is not possible, since we cannot record at the frame rates required without inducing trafficking artefacts due to phototoxicity and drift. Moreover, beta-arrestin sensors, for example those from Montana Molecular, have not yet been tested in the tissue setting. Nonetheless, we can be reasonably sure that differing second messenger generation is likely involved in the observed heterogeneity, since each agonist has a different signal bias with regard to beta-arrestin, cAMP and Ca²⁺, hence why they were studied. We now discuss the need for directly correlative trafficking/cAMP/beta-arrestin measures across the pancreatic islet, for example using lightsheet microscopy, as follows (lines 394-397):

“Fourthly, trafficking was not correlated with second messenger generation or beta-arrestin recruitment at the individual cell level. Given the nature of the trafficking experiments performed here (3D multicellular timelapse) such multiparametric experiments are challenging and will require lightsheet imaging to achieve the frame rates required.”

The authors hint at differential tissues in their discussion. Given the controversies in tissue-specific expression of GLP1R, can the authors comment on the tissue distribution of the receptor in this mouse model?

Thanks for touching upon this. It is exactly because of the lack of specific, validated reagents that we decided to generate the SNAP-GLP1R mice. We and others were fed up of trying and failing to follow up spurious findings (see our review here on GLP1R/GIPR detection ¹). Our previous methodology for detecting GLP1R, the LUXendins ^{2,3}, were requested by dozens of labs worldwide, who are now using these reagents to provide new insight into GLP1R biology. For example, Pauza et al. recently showed that GLP1R is in the carotid body, which is responsible for the hemodynamic effects of GLP1RA. Along similar lines, Cahill et al. recently showed that GLP1R regulates thromboxane-induced human platelet activation.

We expect GLP1R^{SNAP/SNAP} mice to drive similar advances in GLP1R biology and to this end have already distributed animals under MTA to various labs. We are aware that a number of investigators will use these mice to report on GLP1R expression/signaling in their respective tissue. Without wanting to step on the feet of our more eminently qualified colleagues, we now expand the discussion to acknowledge that SNAP-GLP1R mice provide a step-change in terms of detecting GLP1R protein expression, as follows (lines 397-404):

“Lastly, we focused our efforts on pancreatic islets as a tractable and relevant testbed for endogenous GPCR detection. However, GLP1RA and GLP1R/GIPR dual agonists, approved for the treatment of type 2 diabetes and more latterly obesity, have also shown efficacy for neurodegenerative and inflammatory disease states ⁴. Yet localizing GLP1R protein and

assigning cellular/molecular targets remains challenging due to lack of specific and sensitive reagents ¹. We thus expect GLP1R^{SNAP/SNAP} mice and accompanying SNAP labels to inform tissue- and cell-specific GLP1R protein expression patterns.”

Overall, the methodologies utilised are cutting edge and appropriate for the research questions and study well controlled and carried out.

Thank you again for the feedback.

Reviewer #2 (Remarks to the Author):

Ast et al. created a mouse with a SNAP-tag knocked into the N-terminus of the GLP1 receptor. The authors tested different dyes in pancreatic islets, an example for primary tissue staining. Different tests and stimulations show that the knock-in is functional. The testing is basically well done, but there are several points where details are missing or the explanation should be expanded. To be able to classify the quality and specificity of the labelling I had difficulties to see single cells in the small images. The images could be easily expanded showing magnifications of single cells so that the amount of cytosolic to membrane labeling could be estimated better. This could be shown in the supplement.

We thank the reviewer for their time and positive assessment of our manuscript. As suggested, we have provided magnifications of the images to better show SNAP-GLP1R labelling in single cells (Figure 2b and e, Figure 3b and c, Figure 4b, Supplementary Figure 2a and b). We apologize that this was not clear in the original submission.

It is also not clear to me why there is such a large amount of cytosolic label; for example, in Fig. 2b, d and e, the labeling with BG-TMR seems to be strong in the cytosol and not just in the membrane. I guess the dark spots are the nuclei. Why is this so strong? Is this specific labeling of internal receptor or unspecific background? Would more washing steps help? Is this as strong in single, dissociated cells or only in the tissue?

BG-TMR labeling is specific, since signal is almost absent in GLP1R^{WT/WT} islets, and as the reviewer notices, there is no GLP1R in the nucleus, which remain as darker spots. Surface and cytoplasmic labelling are expected, since GLP1R is present in the cytoplasm due to constitutive activity (shown to be ~25% in Figure 5c), as well trafficking of nascent receptor from the endoplasmic reticulum ⁵. It is for this reason that we generated cell non-permeable sulfonated dyes, allowing interrogation of the surface GLP1R pool. While the reviewer makes a good suggestion, more washing only serves to strip both membrane and cytosolic staining. We have however performed further experiments in dissociated beta cells and see that BG-TMR similarly labels the cell surface and cytoplasm, whereas SBG-TMR is more surface-restricted (Supplementary Figure 4a). We also show that BG-TMR and GLP1R mAb staining are highly co-localized as shown by the Mander's co-efficient of 0.958 (lines 168-170). Together, these experiments further show that BG-TMR labelling is specific and that cytoplasmic staining is not a tissue-dependent phenomenon. Our data thus further reinforce the rationale for validating permeable and non-permeable SNAP dyes in the studies here.

List of comments:

- Halo-tags might be better than SNAP-tags for far-red rhodamines as shown in <https://doi.org/10.1016/j.chembiol.2019.01.003> This should be at least discussed or tested in cell culture.

Since we are using many different dyes and imaging approaches, it is difficult to *a priori* know which tag to incorporate. We selected the SNAP-tag over the Halo-tag for the following reasons: 1) the Cisbio SNAP-GLP1R construct is the gold-standard for pharmacological characterization of GLP1RA and is known to signal and traffic identically to native GLP1R, whereas the Halo-GLP1R has not been subject to the same level of investigation; 2) the SNAP-tag is significantly smaller (20 kDa) compared to Halo-tag (30 kDa), which in turn makes CRISPR/Cas9 gene engineering more straightforward, since the provided repair template is smaller in size and less prone to off-target incorporation; 3) unlike the Halo-tag, dyes linked to the SNAP-tag remain after chemical fixation (see Figure 2f and g), allowing multiplexed antibody staining, as well as opening up super-resolution approaches such as STORM that do not work well in live tissue; and 4) ultimately, SNAP-tag is orthogonal to Halo-tag and using either protein self-label opens up the potential to visualize two proteins. On the flip side Halo-tag is better for visualization of rhodamine-based dyes, as elegantly shown by Erdmann et al ⁶ and confirmed by ourselves ⁷. We have now expanded the discussion to incorporate the rationale for using SNAP-tag versus Halo-tag, as follows (lines 339-344):

“For the studies here, we selected SNAP- over Halo-tag, since the SNAP_hGLP1R construct, used as the basis for the SNAP_F-mGLP1R template, is well characterized in terms of its pharmacology and trafficking over hundreds of studies. We note however that Halo-tags are advantageous in terms of rhodamine dye labeling and STED nanoscopy ^{6,7}. Ultimately, SNAP-tag is orthogonal to Halo-tag and using either protein self-label opens up the potential to simultaneously visualize two proteins (and more with CLIP-tag).“

- Is the labeling the same throughout the whole islet? To show that the substrate penetrates the whole islet a profile through the whole islet should be shown or several images at different depth in the supplement.

Islets are large micro-organs (~ 150-200 μm) packed with highly-scattering secretory granules. As such, it is not possible to penetrate the full islet depth using conventional microscopy. While multiphoton imaging is helpful in this regard, many of the fluorophores used here have poor or unpredictable two photon excitation cross-section ⁸. As such, we now provide confocal z-stacks showing bright GLP1R^{SNAP/SNAP} labelling at 50 μm islet depth within the islet core (Supplementary Figure 1d). There are no apparent differences in labelling at the islet surface or core (Supplementary Figure 1e). From this, we can conclude that SNAP dyes are (impressively) able to penetrate deep within the tissue *in vitro*.

- The comparison with LUX645 (Fig. 2e) should be shown on much higher magnification (single cells) to see if it labels the same structure.

We agree with the reviewer and have included magnifications of the LUXendin and mAb stainings in the main manuscript (Figure 1b, e, f and g, Figure 4c and d) and in the Supporting Information (Supplementary Figure 2a and b).

- To me it is not clear what the antibody labels in Fig. 2e., it seems to be everywhere. As pointed out above, this needs to be magnified. Both images, BG-JF549 and mAb are in mixed colors, this is difficult. At least a color map is needed.

For successful mAb staining, islets need to be PFA-fixed then permeabilized, which reduces SNAP-label intensity and increases background noise. Nonetheless, the mAb has been validated by us and others using GLP1R KO tissue and is known specific (in fact one of only two available, see review by us¹) (see Figure 3a from <https://doi.org/10.1038/s41467-020-14309-w>). The SNAP-labels are shown to be specific using GLP1R^{WT/WT} littermate control islets (Figure 2a, Figure 3b and c, Figure 4b), and we now show that each label can be blocked using BG-block to prior occupy the SNAP-tag (Figure 2e, Figure 4e and f, Supplementary Figure 2b, Supplementary Figure 4b).

We however agree with the reviewer that the co-staining studies could be clearer. As such, we have repeated experiments using BG-TMR (Figure 2f), BG-JF₅₄₉ (Figure 2g) and BG-SulfoCy5 (Figure 4c) together with mAb to better show antibody staining at the cell surface and cytosol. We also provide magnified images, have omitted DAPI for clarity, use red and cyan instead of red and green, and show separate panels for each stain to better show co-localization. Lastly, we have provided quantification of co-localization between SNAP-labels and GLP1R mAb, in each case showing a high Mander's co-efficient (and hence strong co-localization) (Mander's co-efficient for overlap with mAb: BG-TMR = 0.958 ± 0.037 , BG-JF₅₄₉ = 0.970 ± 0.064 , SulfoCy5 = 0.932 ± 0.044 ; mean \pm S.D).

- The magnifications in Fig. 2b, 3b, 3e and others should show a scale bar.

Thanks for pointing this omission out – scale bars are now included.

- Figure 3 legend: “SBG-TMR and BG... lead to brighter and cleaner labeling...”. This should be shown and explained. The brighter labeling should be shown, for example, with a scale bar of the fluorescence signal. I would not expect that the labeling of receptors in the membrane is different between both substrates (with and without the sulfonate). If it is different, is this due to the labeling efficacy or the quantum yield of the fluorophore? However, the latter should not change by modifying the BG linker.

The reviewer touches on an important point. First, we have split Figure 3 into two new figures to highlight magnification panels (Figures 3 and 4). Second, we have measured the signal intensities with line scans and as expected observe more surface-restricted labelling and increased brightness for SBG/Sulfo versus BG-dyes (Figure 3d). We attribute this to the fact that more dye is available for surface labelling, since it is not ‘consumed’ by receptors residing in intracellular stores (vide supra). The reviewer is therefore correct that this has to do with labelling efficiency and not with QY or any other photophysical dye properties.

- “Cleaner signal” such as mentioned in line 135 or “cleaner labeling” in line 680 is not a good scientific term and should be better explained /defined.
- On the same line, “... more specific than ...” in line 136 should be better explained.

Thanks for pointing this out – both now rectified.

- Restricting the labelling to the surface is in principle good, but it needs to be clear that it is specific. Why not performing dSTORM also with the LUXendins? Or the mAbs?

One motivation for performing dSTORM with a knock-in SNAP-tag is to be able to look at the non-stimulated receptor in the tissue setting. This is not possible with LUXendins, as they bind the orthosteric site and antagonize the receptor, potentially altering its movement and

localization. Primary and secondary mAbs on the other hand add a huge bulk to the receptor, and outsource the fluorophore tens of nanometers, thereby decreasing the gains in resolution beyond the diffraction limit. Combining either LUXendin or mAb with SNAP dye for dSTORM is complicated, since Alexa647 secondaries show poor labelling even in fixed tissue and LUXendin is either Cy5-based (i.e. non-orthogonal) or non-photoblinking.

Nonetheless, the observed surface GLP1R labeling/localization is specific, since 1) SBG- and Sulfo-dyes do not label GLP1R^{WT/WT} islets (Figure 3 and Figure 4); and 2) GLP1R^{WT/WT} islets labeled with BG-SulfoCy5 do not generate detectable dSTORM signal (Figure 5). These are gold-standard controls for validation of dye and antibody specificity. We now also provide further experiments showing that BG-SulfoCy5 labeling can be blocked using BG-Block to occupy the SNAP-tag, and that BG-SulfoCy5 co-localizes with GLP1R mAb as well as LUX551 at the cell surface (Cy3-based GLP1R antagonist) (Figure 2g and h). Lastly, we have calculated clusters/ μm^2 and find a marked increase in their number in GLP1R^{SNAP/SNAP} versus GLP1R^{WT/WT} islets (lines 289-295).

- DBSCAN should be explained in 1-2 sentences.

We agree and have added 1-2 sentences of explanation.

- Fig. 6c: The colour map is red-white, DAPI in white would be difficult. I am not sure if there is a DAPI labeling in these images? Scale bar should be added to the zoom images.

The DAPI panel was an error on our behalf and has been removed. Scale bars were added.

- Fig. 6e: Are both images showing the same cell?

Apologies that this was unclear. It is difficult to compare basal and stimulated conditions in the same cell, as laser exposure photobleaches the dye, so the 2nd recording would have fewer molecules. Thus, the image shows two different cells, allowing comparison of GLP1R trajectories over the same timecourse and laser exposure. We have now clarified this in the Material and Methods, as well as the Results as follows (lines 561-564):

“Since laser exposure leads to non-linear photobleaching, basal and stimulated conditions were studied in separate cells to allow comparison of GLP1R trajectories over the same time course and laser exposure.”

- Line 163-165: Verb missing.

The sentence has been corrected as follows: “Together, these data show that sulfonated SNAP-dyes label the cell surface, with particular application to the visualization of endogenous proteins in complex tissue.”

- Line 391: “Detection” instead of “emission”. The fluorescence emission is broader.

Thanks for bringing this to our attention – now rectified.

Reviewer #3 (Remarks to the Author):

The manuscript of Ast and colleagues reports a new mouse model in which GLP-1 receptor expression and signaling can be monitored. The authors use a SNAP-tagging

approach and report validation studies as well as an initial pass of characterization of the expression and tracking of GLP-1 receptors in tissues including pancreatic islets. Complex responses are observed, both with respect to heterogenous cell types, as well as specific identified GLP-1R agonists. The studies are well thought out, and appropriate controls are utilized and presented.

We thank the reviewer for their kind and constructive comments.

I offer the following suggestions for improvement:

1. The results section is written quite superficially throughout. Statistical findings are not presented beyond occasional statement of whether something was statistically significant or not. Complex panels of graphs are only vaguely described. As just one example, Figure 1b and 1c present crucial validation studies but are described in the text only as “The construct was initially tested in vitro and found to signal and localize identically to human GLP1R-GFP and SNAP-GLP1R constructs widely used in cell biology applications (Figure 1b and c).” To be clear, this is just a single example; the results section should be thoroughly re-written to describe the presented data in much greater detail. Adding specific statistical results into figure legends similar to what the group presented in Ast et al., Nat Commun. 2020 Jan 24;11(1):467. doi: 10.1038/s41467-020-14309-w would also be very helpful in this regard.

The manuscript was originally written as a shorter article for another NPG journal before being transferred directly to *Nature Communications* at the Editor's suggestion. Therefore, the results sections is more 'snappy' than we would have liked. As suggested by the reviewer, we have now re-written the results section to provide a better description of the data, as well as inserted relevant statistical results into the text (e.g. EC_{50} values, Mander's coefficient, trafficking parameters). While statistical tests and n numbers were provided in the original figure legends, we have now revised these in accordance with *Nature Communications* requirements, including insertion of dF and F-test parameters.

2. Statistical tests are defined well in Methods, however it is unclear in what cases replicates are based in something other than an individual animal. In some cases it appears that replicates were treated as something else, for example an individual pancreatic islet and/or culture preparation rather than an animal. Please make clear, rigorous and transparent and ensure that in vivo/ex vivo data are based on a mouse = n of 1.

The reviewer raises an important point. All studies in the UK involving protected species require a statistician sign-off as part of the ethical approval process and the current manuscript is no exception. In terms of defining the replicate as well as presenting data, we referred to the *Journal of Cell Biology* and *Partnership for Assessment and Accreditation of Scientific Practice* statistical reporting guidelines. Both guidelines suggest that: 1) combining data from multiple independent experiments squanders useful information about variability; 2) averaging cell-level data before statistical analysis risks losing potentially important information; 3) no measurement should be excluded from analysis unless the operator is confident that there will be no effect on analyses; and 4) authors should be transparent about data collection, experimental unit, replicate nature, as well as analysis. We now expand upon this in the statistical analysis section, as follows:

“The independent replicate is isolation of islets from a single animal. In all cases, experiments use islets from at least three animals, taken from at least two separate isolation procedures, which is a nuisance variable and hence used as a blocking factor. Since variation between individual islets/cells is greater than variation of the mean between animals, they are considered as separate datapoints for analysis purposes rather than being averaged. Statistical test and accompanying information are provided for each graph in the figure legend.”

3. Recommend moving the Exendin9 data from Suppl Figure (esp panels Suppl Figure 2a and 2d) into main Figure 4. These data are very interesting re: constitutive activity and would be best presented and interpreted with direct comparison to the agonists.

We agree and have shifted the Exendin9 data from the Supplementary Information to Figure 5. Please note that we have also expanded the analysis to include cumulative cell number, internalization strength and internalization rate, to allow easy cross-comparison with vehicle, exendin4, semaglutide and tirzepatide. This interesting result further demonstrates the need to study GPCR signalling in tissue where paracrine inputs are present.

REFERENCES

1. Ast, J., Broichhagen, J. & Hodson, D.J. Reagents and models for detecting endogenous GLP1R and GIPR. *EBioMedicine* **74**, 103739 (2021).
2. Ast, J. *et al.* An expanded LUXendin color palette for GLP1R detection and visualization in vitro and in vivo. (2021).
3. Ast, J. *et al.* Super-resolution microscopy compatible fluorescent probes reveal endogenous glucagon-like peptide-1 receptor distribution and dynamics. *Nat Commun* **11**, 467 (2020).
4. McLean, B.A. *et al.* Revisiting the Complexity of GLP-1 Action from Sites of Synthesis to Receptor Activation. *Endocr Rev* **42**, 101-132 (2021).
5. Poc, P. *et al.* Interrogating surface versus intracellular transmembrane receptor populations using cell-impermeable SNAP-tag substrates. *Chemical Science* **11**, 7871-7883 (2020).
6. Erdmann, R.S. *et al.* Labeling Strategies Matter for Super-Resolution Microscopy: A Comparison between HaloTags and SNAP-tags. *Cell Chemical Biology* **26**, 584-592.e586 (2019).
7. Birke, R. *et al.* Sulfonated red and far-red rhodamines to visualize SNAP- and Halo-tagged cell surface proteins. *Organic & Biomolecular Chemistry* (2022).
8. Grimm, J.B. *et al.* A general method to optimize and functionalize red-shifted rhodamine dyes. *Nature Methods* **17**, 815-821 (2020).

REVIEWERS' COMMENTS

Reviewer #1 (Remarks to the Author):

All comments have been addressed and manuscript improved. There are no additional points to address.

Reviewer #2 (Remarks to the Author):

The authors have addressed all my concerns and I have no further comments or suggestions.

Reviewer #3 (Remarks to the Author):

The authors have fully responded to the original critiques and have significantly improved their paper. I have no further edits to recommend.